# Surrogate Prompt Learning: Towards Efficient and Diverse Prompt Learning for Vision-Language Models

**Liangchen Liu** [1]  **Nannan Wang** [1]  **Xi Yang** [1]  **Xinbo Gao** [1]  **Tongliang Liu** [2]

## Abstract

Prompt learning is a cutting-edge parameter-efficient fine-tuning technique for pre-trained vision-language models (VLMs). Instead of learning a single text prompt, recent works have revealed that learning diverse text prompts can effectively boost the performances on downstream tasks, as the diverse prompted text features can comprehensively depict the visual concepts from different perspectives. However, diverse prompt learning demands enormous computational resources. This efficiency issue still remains unexplored. To achieve efficient and diverse prompt learning, this paper proposes a novel **Surrogate Prompt Learning (SurPL)** framework. Instead of learning diverse text prompts, SurPL directly generates the desired prompted text features via a lightweight **Surrogate Feature Generator (SFG)**, thereby avoiding the complex gradient computation procedure of conventional diverse prompt learning. Concretely, based on a basic prompted text feature, SFG can directly and efficiently generate diverse prompted features according to different pre-defined conditional signals. Extensive experiments indicate the effectiveness of the surrogate prompted text features, and show compelling performances and efficiency of SurPL on various benchmarks. Code is available at `https://github.com/llcllc1997/SurPL`.

## 1. Introduction

Recently, vision-language models (VLMs) (Radford et al., 2021; Jia et al., 2021; Yao et al., 2021; Zhai et al., 2022) have shown great capability in open-world visual under-

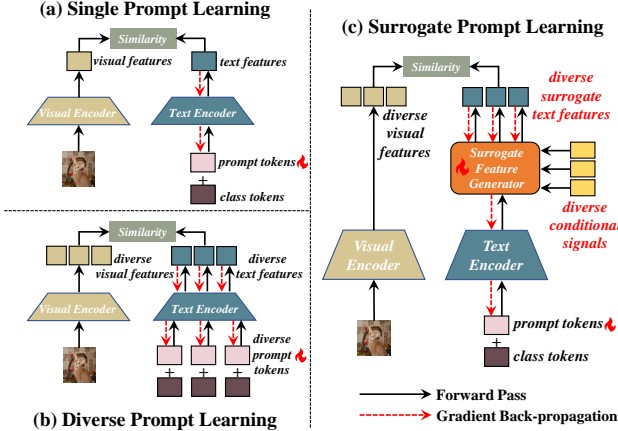

*Figure 1.* Illustration of different prompt learning frameworks. (a): conventional single prompt learning. (b): diverse prompt learning, e.g., instance-dependent prompt learning and fine-grained prompt learning, etc. (c) our proposed Surrogate Prompt Learning.

standing. How to effectively and efficiently leverage these large-scale pre-trained models for downstream tasks has garnered significant attention in research. In this context, prompt learning (Zhou et al., 2022c;b; Chen et al., 2023; Khattak et al., 2023a;b; Roy & Etemad, 2024; Yao et al., 2024; Lafon et al., 2024; Zhang et al., 2024a) comes to the fore as an emerging parameter-efficient fine-tuning (PEFT) technique, and has achieved compelling transferring performances for pre-trained VLMs.

Given a downstream task, prompt learning targets to learn a few extra prompt tokens at the text input position, so that the prompted text feature can obtain the task-specific knowledge, and provide more accurate descriptions of the visual concepts in the downstream tasks. While previous methods mainly focus on learning one single text prompt, recent approaches start to learn diverse text prompts and achieve significant improvements. These diverse prompted text features constitute more comprehensive descriptions of the visual concepts in downstream tasks from different perspectives. For example, instance-dependent prompt learning (Zhou et al., 2022b) learns a unique visual-guided text prompt for each visual example; fine-grained prompt learning (Chen

---

[1]Xidian University, Xi'an 710071, Shaanxi, China. [2]The University of Sydney, Darlington, NSW 2008, Australia. Correspondence to: Nannan Wang <nnwang@xidian.edu.cn>.

*Proceedings of the 42$^{nd}$ International Conference on Machine Learning*, Vancouver, Canada. PMLR 267, 2025. Copyright 2025 by the author(s).

et al., 2023; Wang et al., 2023; Lafon et al., 2024) learns several text prompts to align with different local-based visual features, etc. We collectively denote these methods as diverse prompt learning.

However, diverse prompt learning demands enormous computational resources. Although prompt learning is parameter-efficient, optimizing the prompt tokens at the input position still requires back-propagated gradient computation through the entire text encoder, which causes considerable consumption on GPU memory and optimization duration. Given an $M$-class visual classification task, learning a single text prompt needs to process the back-propagated gradient computation from $M$ prompted text features to the input, simultaneously (shown in Fig.1 (a)). Nevertheless, for instance-dependent prompt learning, the optimization involves gradient computation w.r.t $B \times M$ text features, where $B$ is the training batch size of the visual samples. For fine-grained prompt learning, the gradient computation will increase to $Z \times M$ text features, where $Z$ indicates the number of applied local visual features. As illustrated in Fig.1 (b), these existing diverse prompt learning methods scale up the computation consumption from $\mathcal{O}(M)$ to $\mathcal{O}(B \times M)$ or $\mathcal{O}(Z \times M)$, thereby leading to severe efficiency issue.

To breakthrough the dilemma, this paper proposes **Surrogate Prompt Learning (SurPL)**. Instead of learning diverse text prompts from scratch, SurPL directly generates their prompted text features via a lightweight **Surrogate Feature Generator (SFG)**. As shown in Fig.1 (c), SurPL only learns a single basic prompt. Given the basic prompted text feature, SFG can efficiently generate diverse text features according to different conditional signals. These signals are inherently flexible and arbitrary. They can be pre-defined prior knowledge (e.g., semantic information of visual instances), or can be initialized and learnable tokens that are supposed to convey specific knowledge (e.g., fine-grained information). **SurPL bypasses the issue of enormous gradient computation inside the text encoder** (i.e., keeping the computation consumption under $\mathcal{O}(M)$), but still provides diverse text features that can be exploited to comprehensively align with visual concepts.

**We emphasize that the core idea of this paper is not designing a new diverse prompt learning method, but proposing a novel and unified framework that can implement arbitrary diverse prompt learning approaches under the computation efficiency that is comparable to single prompt learning approaches.** Qualitative and quantitative experiments have verified that our generated surrogate features are sufficiently effective to replace those original prompted features. We further show that by simply incorporating both instance-dependent and fine-grained prompt learning concepts into our surrogate prompt learning framework, SurPL can even surpass the state-of-the-art

performances of existing diverse prompt learning methods, while maintaining remarkable computation efficiency.

In a nutshell, we present our contributions as follows: 1) We propose a novel and efficient paradigm for diverse prompt learning, denoted as Surrogate Prompt Learning (SurPL). SurPL directly generates the diverse prompted text features instead of learning diverse text prompts from scratch, addressing the efficiency issue in existing diverse prompt learning; 2) To generate diverse text features, we propose a Surrogate Feature Generator (SFG). These features are controlled and generated by flexible and arbitrary conditional signals, allowing SurPL to adapt and integrate different types of diverse prompt learning frameworks; 3) Extensive experiments are conducted to indicate the effectiveness of our generated surrogate text features, and show remarkable performances and efficiency of SurPL on various benchmarks.

## 2. Related Works

**Single prompt learning:** Recently, prompt learning has become a popular parameter-efficient fine-tuning technique that can rapidly adapt pre-trained VLMs (e.g., CLIP (Radford et al., 2021), ALIGN (Jia et al., 2021) and LiT (Zhai et al., 2022)) to various downstream tasks, without tuning any pre-trained parameters. Pioneer work CoOp (Zhou et al., 2022c) replaces hand-crafted prompt templates by a set of learnable prompt tokens at the text input position. After learning on downstream tasks, these prompt tokens capture task-specific knowledge and provide more accurate prompted text features to align with task-related visual concepts, thus exhibiting better performances. Inspired by CoOp, subsequent approaches dedicate to further improve VLM's transfer ability by learning multi-modal prompts (Khattak et al., 2023a; Xu et al., 2023; Khattak et al., 2023b; Cho et al., 2023; Wang et al., 2023), generalizable prompts (Zhu et al., 2023; Yao et al., 2023; Liu et al., 2023; Yao et al., 2024; Zhang et al., 2024a), meta learning-based prompts (Li et al., 2023a; Zhao et al., 2024), neural architecture search-based prompts (Zhang et al., 2024b), etc.

**Diverse prompt learning:** Different from the above approaches that only learn prompts to obtain a single prompted text feature, recent studies have manifested that leveraging diverse prompts can significantly improve the transfer performances, as their prompted text features can provide a more comprehensive description of the visual concepts from different perspectives. CoCoOp (Zhou et al., 2022b) proposes instance-dependent prompt learning, i.e., learning dynamic visual-conditional prompt for each visual example. PLOT (Chen et al., 2023) and ALIGN (Wang et al., 2023) learn multiple fine-grained text prompts based on the optimal transport strategy. GalLoP (Lafon et al., 2024) targets to learn both global and local text prompts that can align with the global and local representation features of the visual

concepts. Although these diverse prompt learning methods have achieved remarkable performances, they still demand enormous computation resources.

## 3. Proposed Method

### 3.1. Overview

This paper proposes Surrogate Prompt Learning (SurPL) to achieve efficient and diverse prompt learning. Instead of directly learning diverse text prompts from scratch, SurPL alternatively generates their prompted text features. As shown in Fig.2, SurPL introduces a novel Surrogate Feature Generator (SFG), which can generate diverse text features based on the obtained basic text feature according to different conditional signals. These signals are inherently flexible and arbitrary, enabling SurPL to incorporate any type of diverse prompt learning framework.

In this work, we focus on two primary types of diverse prompt learning methods: instance-dependent and fine-grained prompt learning. Similar to existing single prompt learning approaches, SurPL only optimizes a few text prompt tokens to obtain a basic prompted text feature. Using this basic text feature and various pre-defined conditional signals, SurPL efficiently generates instance-dependent and fine-grained text features via SFG, which are supposed to be obtained by learning diverse text prompts. These generated text features, combined with the global-invariant text feature, provide more comprehensive descriptions of the visual concepts from different perspectives, thereby significantly boosting the adaptation ability of VLMs.

### 3.2. Preliminaries

**Contrastive Language-Image Pre-training (CLIP).** CLIP (Radford et al., 2021) has shown great potential on zero-shot image classification. CLIP consists of a visual-encoder $\mathcal{V} = \{V^k\}_{k=1}^K$ and a text encoder $\mathcal{T} = \{T^k\}_{k=1}^K$, where $K$ denotes the depth of the encoder layers. Given an $M$-class task, the text inputs $\boldsymbol{t} = \{\boldsymbol{t}_m\}_{m=1}^M$ are formed by the combination of the classnames (text labels) and the hand-crafted prompt, e.g., 'a photo of a [classname]'. The output text features can be obtained as $\boldsymbol{w} = \mathcal{T}(\boldsymbol{t}) = \{\boldsymbol{w}_m\}_{m=1}^M$., where $\boldsymbol{w}_m$ denotes the feature corresponds to $m$-th class. Given the visual feature of an image $\boldsymbol{x}$: $\hat{\boldsymbol{f}} = \mathcal{V}(\boldsymbol{x})$, the prediction probability of $\boldsymbol{x}$ belonging to $m$-th class can be expressed as:

$$p(m|\boldsymbol{x}) = \frac{\exp(\cos(\hat{\boldsymbol{f}}, \boldsymbol{w}_m)/\tau)}{\sum_{j=1}^M \exp(\cos(\hat{\boldsymbol{f}}, \boldsymbol{w}_j)/\tau)}, \qquad (1)$$

where $\cos(\cdot, \cdot)$ denotes the cosine similarity and $\tau$ is a temperature parameter learned by CLIP.

**Context Optimization (CoOp).** CoOp (Zhou et al., 2022c)

replaces the hand-crafted prompts by several learnable prompt tokens $\boldsymbol{s} = \{\boldsymbol{s}_l\}_{l=1}^L \in \mathbb{R}^{L \times d}$ at the text input position, where $L$ and $d$ indicates the prompt token length and token dimension. Therefore, the text input of $m$-th class can be formed as $\boldsymbol{t}_m = [\boldsymbol{s}, \boldsymbol{c}_m]$, where $\boldsymbol{c}_m$ is the $m$-th classname. Given a visual sample $\boldsymbol{x}$ and the ground-truth label $y_m$ of the downstream task, CoOp optimizes prompt token parameters $\boldsymbol{s}$ via cross-entropy loss $\mathcal{L}_{CE}$ between the prediction probability and label (CLIP parameters are fixed):

$$\mathcal{L}_{CE}(\boldsymbol{s}) = -\sum_m y_m \log p(m|\boldsymbol{x}),$$
$$\text{where} \quad p(m|\boldsymbol{x}) = \frac{\exp(\cos(\hat{\boldsymbol{f}}, \boldsymbol{w}_m)/\tau)}{\sum_{j=1}^M \exp(\cos(\hat{\boldsymbol{f}}, \boldsymbol{w}_j)/\tau)}, \qquad (2)$$

**Dense Visual-Text Prompt (DVLP).** DVLP is a stronger baseline that has been widely-used in recent researches (Khattak et al., 2023a;b; Zhang et al., 2024a). DVLP exploits multi-modal dense prompt tokens at each encoder layer. DVLP consists of text prompts $\boldsymbol{s} = \{\boldsymbol{s}^k\}_{k=1}^K$ and visual prompts $\boldsymbol{u} = \{\boldsymbol{u}^k\}_{k=1}^K$, where $K$ denotes the depth of prompts. The computation procedure of the $k$-th layer of DVLP can be expressed as:

$$\begin{cases} [\boldsymbol{s}^{k+1}, \boldsymbol{c}_j^{k+1}]_{j=1}^M = T^k([\boldsymbol{s}^k, \boldsymbol{c}_j^k]_{j=1}^M), \\ [\boldsymbol{x}^{k+1}, \boldsymbol{u}^{k+1}] = V^k([\boldsymbol{x}^k, \boldsymbol{u}^k]). \end{cases} \qquad (3)$$

The output of $\boldsymbol{s}^k$ and $\boldsymbol{u}^k$ are then substituted by the newly involved prompt tokens $\boldsymbol{s}^{k+1}$ and $\boldsymbol{u}^{k+1}$, and reformed as the input for the next $k + 1$-th layer.

### 3.3. Surrogate Feature Generator

As shown in Fig.2, the proposed Surrogate Feature Generator (SFG) is intrinsically a cross-attention module. The "query" is the basic prompted text feature while the "key" and "value" are the pre-defined conditional signals. Different from standard cross-attention modules, we decrease the hidden layer dimension of the MLP (Multi-Layer Perceptron), and keep SFG still parameter-efficient.

We denote $\boldsymbol{w} = \{\boldsymbol{w}_m\}_{m=1}^M \in \mathbb{R}^{M \times d}$ as the basic prompted text feature obtained from the text encoder, where $M$ and $d$ indicate the number of classes and the feature dimension. Generally, $\boldsymbol{w}$ can be obtained by any single prompt learning method. In this paper, we utilize DVLP as the baseline. The conditional signal $\boldsymbol{\alpha} \in \mathbb{R}^d$ is flexible and arbitrary, which can be any pre-defined prior knowledge (e.g., semantic information of visual instances), or initialized and learnable tokens that are supposed to convey specific knowledge (e.g., fine-grained information). The generated surrogate text fea-

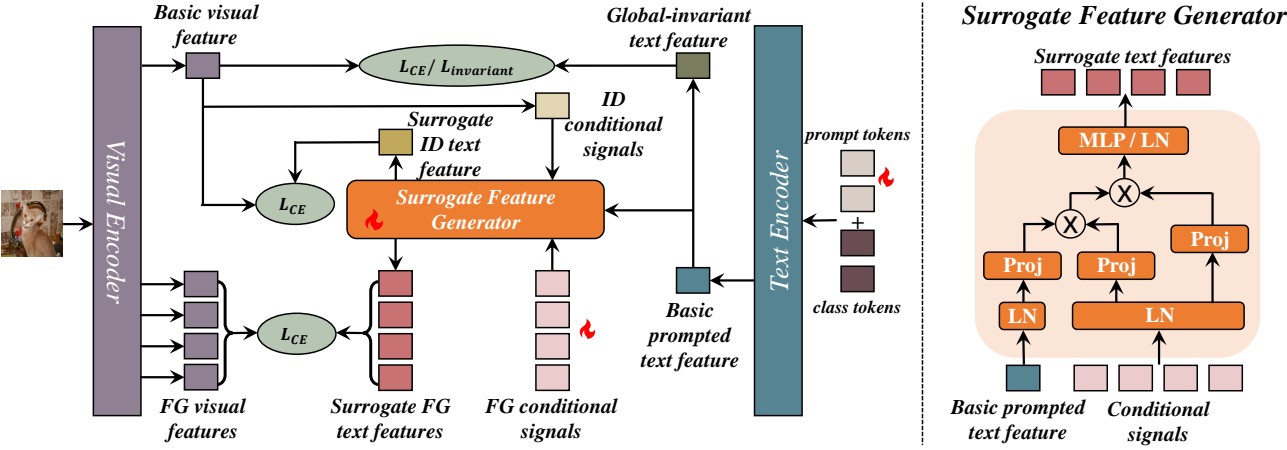

*Figure 2.* Pipeline of our proposed Surrogate Prompt Learning (SurPL). In this work, we mainly generate two types of diverse prompted text features: instance-dependent features and fine-grained features. Notably, "**ID**" and "**FG**" denote "**instance-dependent**" and "**fine-grained**" in the figure, respectively.

ture $\boldsymbol{h} \in \mathbb{R}^{M \times d}$ corresponding to $\boldsymbol{\alpha}$ can be derived as:

$$\boldsymbol{h} = \boldsymbol{\theta}_{\text{SFG}}(\boldsymbol{w}, \boldsymbol{\alpha}) = MLP\left(softmax\left(\frac{\boldsymbol{w}\boldsymbol{\alpha}^{\top}}{\sqrt{d_k}}\right)\boldsymbol{\alpha}\right), \tag{4}$$

where $\boldsymbol{\theta}_{\text{SFG}}$ represents the parameters of SFG. For simplification, we omit the necessary feature projection ($Proj$) and layer norm ($LN$) operations in Eq.4, and provide detailed computation procedures in the appendix.

SFG uses conditional signals as guidance, so the generated text feature obtains specific knowledge related to the given signal. Leveraging diverse condition signals can generate diverse text features that comprehensively align with visual concepts in downstream tasks from different perspectives. This operation avoids learning diverse prompts from scratch, hence addressing the efficiency problem in existing diverse prompt learning methods. In the following, we provide the concrete implementations of instance-dependent and fine-grained surrogate prompt learning.

### 3.4. Instance-dependent Surrogate Prompt Learning

Instance-dependent (ID) prompt learning targets to embed visual information into text prompts, so that the prompted text features can dynamically adapt according to the input visual instances. Given a of visual instances $\boldsymbol{X} = \{\boldsymbol{x}_b\}_{b=1}^{B}$ ($B$ denotes the batch size), instance-dependent prompt learning involves $B\times$ computation cost of gradient back-propagation in the text encoder compared to single text prompt learning methods. This implementation drastically decreases the optimization efficiency, especially when using a sufficiently large batch size.

To address this issue, we generate the instance-dependent

prompted text features via SFG. Concretely, for each visual instance $\boldsymbol{x}_b$, we directly adopt the output visual feature $\hat{\boldsymbol{f}}_{\boldsymbol{b}} = \mathcal{V}(\boldsymbol{x}_b)$ as the instance-dependent conditional signal $\boldsymbol{\alpha}_b^{\text{ID}}$. The surrogate instance-dependent (ID) text feature $\boldsymbol{h}_b^{\text{ID}} \in \mathbb{R}^{M \times d}$ can thus be expressed as: $\boldsymbol{h}_b^{\text{ID}} = \boldsymbol{\theta}_{\text{SFG}}(\boldsymbol{w}, \boldsymbol{\alpha}_b^{\text{ID}})$. $\boldsymbol{h}_b^{\text{ID}}$ encapsulates specific knowledge w.r.t $\boldsymbol{x}_b$, thereby providing more accurate description to the visual instance. Finally, the corresponding instance-dependent optimization loss $\mathcal{L}_{CE}^{\text{ID}}$ is computed as:

$$\mathcal{L}_{CE}^{\text{ID}} = \frac{1}{B}\sum_{b=1}^{B}\left(-\sum_{m}y_m \log p_b^{\text{ID}}(m|\boldsymbol{x}_b)\right),$$

$$\text{where} \quad p_b^{\text{ID}}(m|\boldsymbol{x}_b) = \frac{\exp(\cos(\hat{\boldsymbol{f}}_b, \boldsymbol{h}_{b,m}^{\text{ID}})/\tau)}{\sum_{j=1}^{M}\exp(\cos(\hat{\boldsymbol{f}}_b, \boldsymbol{h}_{b,j}^{\text{ID}})/\tau)}. \tag{5}$$

### 3.5. Fine-Grained Surrogate Prompt Learning

The output of a visual example from the pre-trained VLM visual encoder is actually constituted by a basic (global) visual feature $\hat{\boldsymbol{f}} \in \mathbb{R}^d$ and $N$ fine-grained (local) visual features $\boldsymbol{f} = \{\boldsymbol{f}_n\}_{n=0}^{N-1} \in \mathbb{R}^{N \times d}$. While traditional works typically utilize $\hat{\boldsymbol{f}}$ as the representation of visual concepts, recent approaches start to fully exploit effective fine-grained visual features embedded in $\boldsymbol{f}$, and learn additional fine-grained (FG) text prompts to align with them. Suppose the number of newly involved prompts is $Z$. Fine-grained prompt learning increases the computation cost of gradient back-propagation in the text encoder by $Z$ times compared to single text prompt learning methods.

Since fine-grained information does not require prior knowledge, we simply pre-define the fine-grained (FG) conditional signals as learnable parameters: $\boldsymbol{\alpha}^{\text{FG}} \in \mathbb{R}^{Z \times d}$. $\boldsymbol{\alpha}^{\text{FG}}$

can automatically capture fine-grained cues during the optimization procedure. The surrogate fine-grained (FG) text features $h^{\text{FG}} = \{h_z^{\text{FG}}\}_{z=1}^{Z} \in \mathbb{R}^{Z \times M \times d}$ can be expressed as: $h^{\text{FG}} = \theta_{\text{SFG}}(w, \alpha^{\text{FG}})$.

Following GalLoP (Lafon et al., 2024), we excavate effective fine-grained visual features from $f$ by measuring the similarity between $f$ and $h^{\text{FG}}$. Refer to recent works (Sun et al., 2022; Zhou et al., 2022a; Lafon et al., 2024), $f \in \mathbb{R}^{N \times d}$ is obtained by forwarding them through the last visual encoder block, where the self-attention operation is removed and a linear projection layer is added at the end. Concretely, for the $z$-th fine-grained text feature $h_z^{\text{FG}}$, we select the top $z \times \eta$ visual features with the highest similarities to $h_z^{\text{FG}}$ as the representative fine-grained visual information. Here $\eta$ is a constant factor, which enables the selected fine-grained visual features different and multi-scaled for each $h_z^{\text{FG}}$. The corresponding similarity between $h_z^{\text{FG}}$ and $f$ of the $m$-th class is given by average similarities of the top $z \times \eta$ regions:

$$\text{sim}(h_{z,m}^{\text{FG}}, f) = \frac{1}{z \times \eta} \sum_{n=1}^{N} \mathbb{1}_{\text{top-}z\eta}(n) \cdot \cos(h_{z,m}^{\text{FG}}, f_n),$$

$$\text{where} \quad \mathbb{1}_{\text{top-}z\eta}(n) = \begin{cases} 1, & \text{if rank}_n(\cos(h_{z,m}^{\text{FG}}, f_n)) \leq z\eta, \\ 0, & \text{otherwise.} \end{cases}$$

$$(6)$$

Finally, we compute the fine-grained loss $\mathcal{L}_{CE}^{\text{FG}}$ as:

$$\mathcal{L}_{CE}^{\text{FG}} = \frac{1}{Z} \sum_{z=1}^{Z} \left( -\sum_{m} y_m \log p_z^{\text{FG}}(m|x) \right),$$

$$\text{where} \quad p_z^{\text{FG}}(m|x) = \frac{\exp(\text{sim}(h_{z,m}^{\text{FG}}, f)/\tau)}{\sum_{j=1}^{M} \exp(\text{sim}(h_{z,j}^{\text{FG}}, f))/\tau)}.$$

$$(7)$$

### 3.6. Optimization and Inference

As shown in Fig.2, besides the generated surrogate features $h^{\text{ID}}$ and $h^{\text{FG}}$, we also learn a global-invariant (GI) text feature $w^{\text{GI}}$ based on $w$, which can provide the general and global text description. The global-invariant loss can be expressed as:

$$\mathcal{L}_{CE}^{\text{GI}} = \mathcal{L}_{CE} + \mathcal{L}_{invariant}$$
$$= \mathcal{L}_{CE} + \lambda_1 |w^{\text{GI}} - w^{\text{zs}}| + \lambda_2 |\hat{f} - \hat{f}^{\text{zs}}|, \quad (8)$$

where $\mathcal{L}_{CE}$ indicates the cross-entropy loss calculated between $w^{\text{GI}}$ and $\hat{f}$ according to Eq.2, $w^{\text{zs}}$ and $\hat{f}^{\text{zs}}$ denote the text and visual features obtained from zero-shot text and visual encoders without prompts, $\lambda_1$ and $\lambda_2$ are the coefficients. During the optimization phase, the optimizing parameters $\phi$ include the dense text prompts $s$, dense visual prompts $u$, the Surrogate Feature Generator $\theta_{\text{SFG}}$, fine-grained conditional signals $\alpha^{\text{FG}}$ and the linear projection layer $\theta_{\text{Proj}}$ to obtain $f$. The entire loss is summed by:

$$\mathcal{L}_{\phi = \{s, u, \theta_{\text{SFG}}, \alpha^{\text{FG}}, \theta_{\text{Proj}}\}} = \mathcal{L}_{CE}^{\text{GI}} + \mathcal{L}_{CE}^{\text{ID}} + \mathcal{L}_{CE}^{\text{FG}}. \quad (9)$$

During the inference phase, the final prediction probability of $x$ belonging to the $m$-th class is derived as:

$$p(m|x) = \left( p^{\text{GI}}(m|x) + p^{\text{ID}}(m|x) + \frac{1}{Z} \sum_{z=1}^{Z} p_z^{\text{FG}}(m|x) \right)/3,$$

$$(10)$$

where $p^{\text{GI}}(m|x)$ indicates the prediction probability between $w^{\text{GI}}$ and $\hat{f}$.

**Generalization ability of SurPL:** Recent studies begin to explore the generalization ability of the learned prompts, and evaluate their performances on unseen tasks. To this end, we further propose a generalizable version of SurPL, denoted as SurPL-G. As many previous studies (Keskar et al., 2016; Dziugaite & Roy, 2017; Jiang et al., 2019) have manifested that flat loss landscape leads to better generalization ability, SurPL-G leverages Sharpness-aware Minimization (SAM) (Foret et al., 2021) to optimize the parameters $\phi = \{s, u, \theta_{\text{SFG}}, \alpha^{\text{FG}}, \theta_{\text{Proj}}\}$:

$$\mathcal{L}_{\phi}^{SAM} = \mathcal{L}_{\phi + \hat{\epsilon}} = \mathcal{L}_{\phi + \rho \frac{\nabla \mathcal{L}_{\phi}}{\|\nabla \mathcal{L}_{\phi}\|}}, \quad (11)$$

where $\mathcal{L}_{\phi}$ is computed according to Eq.9 and $\rho$ is the perturbation radius. This implementation encourages the neighboring parameters of $\phi$: $(\phi + \epsilon)$ to have uniformly low loss values, thereby promoting a flatter loss landscape and enhancing the generalization ability of SurPL-G. We provide the detailed algorithms of SurPL and SurPL-G in **appendix**.

## 4. Experiments

**Dataset.** To evaluate the effectiveness of SurPL, this paper exploits 15 public available visual classification datasets as downstream tasks, including ImageNet (Deng et al., 2009), Caltech101 (Fei-Fei et al., 2004), OxfordPets (Parkhi et al., 2012), StanfordCars (Krause et al., 2013), Flowers102 (Nilsback & Zisserman, 2008), Food101 (Bossard et al., 2014), FGVCAircraft (Maji et al., 2013), SUN397 (Xiao et al., 2010), DTD (Cimpoi et al., 2014), EuroSAT (Helber et al., 2019), UCF101 (Soomro et al., 2012), ImageNetV2 (Recht et al., 2019), ImageNet-Sketch (Wang et al., 2019), ImageNet-A (Hendrycks et al., 2021b) and ImageNet-R (Hendrycks et al., 2021a). These datasets constitute a comprehensive benchmark.

**Implementation details.** We adopt Dense Visual-Language Prompt (DVLP) as the baseline model. We mainly evaluate our proposed method on 4 types of experiment settings: SurPL on few-shot learning, SurPL-G on base-to-novel generalization, cross-domain and cross-dataset generalization. Most of the implementation details are followed by PSRC (Khattak et al., 2023b). For SurPL and SurPL-G,

*Table 1.* Efficiency comparison between SurPL and existing diverse prompt learning methods, including CoCoOp(Zhou et al., 2022b), PLOT++(Chen et al., 2023), ALIGN(Wang et al., 2023) and GalLoP(Lafon et al., 2024). '**OOM**" indicates that **GPU Out of Memory** on 4× RTX3090 GPUs.

|  | Methods | Memory | Training | Testing | Acc |
|---|---|---|---|---|---|
| EuroSAT | CoCoOp | 7.79G | 0min42s | 1min17s | 73.32 |
|  | PLOT++ | 13.02G | 0min53s | 0min38s | 92.00 |
|  | ALIGN | 14.00G | 0min41s | 1min40s | 90.77 |
|  | GalLoP | 4.69G | 0min28s | 0min24s | 90.10 |
|  | SurPL | 5.60G | 0min22s | 0min22s | 93.92 |
| Aircraft | CoCoOp | OOM | - | - | 31.21 |
|  | PLOT++ | 20.50G | 8mini21s | 0min21s | 46.74 |
|  | ALIGN | 23.37G | 31min23s | 4min43s | 49.99 |
|  | GalLoP | 18.77G | 5min52s | 0min23s | 58.30 |
|  | SurPL | 7.37G | 2min36s | 0min12s | 60.51 |
| ImageNet | CoCoOp | OOM | - | - | 70.83 |
|  | PLOT++ | OOM | - | - | 72.60 |
|  | ALIGN | OOM | - | - | 72.45 |
|  | GalLoP | OOM | - | - | 75.10 |
|  | SurPL | 23.80G | 8min03s | 6min21s | 74.65 |

*Table 2.* Efficiency comparison between SurPL and existing single prompt learning methods, including MaPLe(Khattak et al., 2023a), PSRC(Khattak et al., 2023b) and our baseline model DVLP.

|  | Methods | Memory | Training | Testing | Acc |
|---|---|---|---|---|---|
| EuroSAT | MaPLe | 5.14G | 0min15s | 0min12s | 92.33 |
|  | PSRC | 5.80G | 0min17s | 0min13s | 92.43 |
|  | DVLP | 5.60G | 0min21s | 0min21s | 92.58 |
|  | SurPL | 5.60G | 0min22s | 0min22s | 93.92 |
| Aircraft | MaPLe | 6.90G | 1min31s | 0min10s | 48.40 |
|  | PSRC | 7.65G | 1min43s | 0min10s | 50.83 |
|  | DVLP | 7.33G | 2min29s | 0min12s | 52.88 |
|  | SurPL | 7.37G | 2min36s | 0min12s | 60.51 |
| ImageNet | MaPLe | 22.68G | 5min53s | 4min22s | 72.33 |
|  | PSRC | 23.10G | 6min14s | 4min31s | 73.17 |
|  | DVLP | 23.30G | 7min44s | 5min48s | 72.62 |
|  | SurPL | 23.80G | 8min03s | 6min21s | 74.65 |

we utilize batch size $B = 4$ for base-to-novel generalization, and batch size $B = 32$ for other three settings. Similar to GalLoP (Lafon et al., 2024), the number of fine-grained text features and multi-scale constant factor are set as $Z = 4$ and $\eta = 10$. All experiments are conducted on a single RTX 3090 GPU. All results are obtained with CLIP ViT-B/16 backbone and 16-shot visual samples unless specified. More detailed implementations are given in appendix.

### 4.1. Efficiency Analysis of SurPL

As shown in Tab.1, we compare the computation-efficiency between SurPL and existing diverse prompt learning methods, under three datasets that have different scales: EuroSAT (10 classes), FGVCAircraft (100 classes) and ImageNet (1000 classes). We set the batch size $B = 32$ as default and report the results of GPU memory, training time (1 epoch for ImageNet and 10 epochs for the other two datasets), testing

*Table 3.* Ablation study of SurPL on few-shot setting and SurPL-G on base-to-novel (B2N) setting. 'GI', 'ID' and 'FG" denote leveraging $w^{\mathrm{GI}}$, $h^{\mathrm{ID}}$ and $h^{\mathrm{FG}}$ to describe the visual concepts, respectively. Averaged results on 11 datasets are reported here. Complete experiment results for individual datasets are given in **appendix**. '**HM**" denotes the harmonic mean between base and novel results.

|  |  | DVLP | GI | GI+ID | GI+FG | GI+ID+FG |
|---|---|---|---|---|---|---|
| Few-shot |  | 82.92 | 83.70 | 83.96 | 84.99 | 85.12 |
| B2N | Base | 79.61 | 81.84 | 84.77 | 85.52 | 86.37 |
|  | Novel | 71.63 | 74.68 | 76.19 | 75.50 | 76.32 |
|  | HM | 75.41 | 78.09 | 80.25 | 80.20 | 81.03 |

time and few-shot learning accuracy. Compared to these diverse prompt learning methods, SurPL significantly reduces computational consumption in terms of GPU memory, training time, and testing time, while even simultaneously improving accuracy. These results indicate that SurPL has achieved our primary goal: improving the computation efficiency for diverse prompt learning.

Notably, for large-scale datasets such as ImageNet, we observe that existing diverse prompt learning approaches require extremely enormous GPU memory. We also notice that the efficiency issue cannot be fully solved by parallel multiple GPUs, since all prompted text features should always be repeatedly computed on each GPU. As we estimate, these methods require at least $80 - 100G$ memory for total, and each GPU should afford $40 - 50G$ memory. In contrast, SurPL can conduct experiments on ImageNet with a single 24G RTX 3090 GPU, which significantly improves the computation efficiency.

Furthermore, we compare SurPL with recently proposed single prompt learning methods. As shown in Tab.2, SurPL achieves significant improvements on accuracy, with only a slight increase in training and testing time. Notably, we find that the GPU memory consumption remains comparable to these single prompt learning methods, which highlights the computation efficiency of our proposed SurPL.

### 4.2. Ablation Study

SurPL learns three kinds of text features: global-invariant (GI) text feature $w^{\mathrm{GI}}$, the surrogate instance-dependent (ID) text feature $h^{\mathrm{ID}}$ and the surrogate fine-grained (FG) text feature $h^{\mathrm{FG}}$. These text features jointly provide a comprehensive description of the visual concept, thereby achieving compelling performance improvements over the baseline DVLP. We conduct ablation study to verify the effectiveness of $w^{\mathrm{GI}}$, $h^{\mathrm{ID}}$ and $h^{\mathrm{FG}}$ under both few-shot learning and base-to-novel generalization settings.

As shown in Tab.3, each of the applied text features ($w^{\mathrm{GI}}$, $h^{\mathrm{ID}}$ and $h^{\mathrm{FG}}$) consistently improves the performance under both experiment settings. These results verify that learning

*Table 4.* Few-shot learning performance comparison between our proposed SurPL and existing single prompt learning (CoOp(Zhou et al., 2022c), MaPLe(Khattak et al., 2023a), DAPT(Cho et al., 2023), PSRC(Khattak et al., 2023b) and LLaMP(Zheng et al., 2024)) and diverse prompt learning (CoCoOp(Zhou et al., 2022b), PLOT++(Chen et al., 2023), ALIGN(Wang et al., 2023) and GaLoP(Lafon et al., 2024)) approaches. The **bold** numbers and underlined numbers denote the best and second-best results, respectively.

| | Methods | ImageNet | Caltech101 | Pets | Cars | Flowers | Food101 | Aircraft | Sun397 | DTD | EuroSAT | UCF101 | AVG |
|---|---|---|---|---|---|---|---|---|---|---|---|---|---|
| **Single** | CoOp | 71.87 | 95.57 | 91.87 | 83.07 | 97.07 | 84.20 | 43.40 | 74.67 | 69.87 | 84.93 | 82.23 | 79.89 |
| | MaPLe | 72.33 | 96.00 | 92.83 | 83.57 | 97.00 | 85.33 | 48.40 | 75.53 | 71.33 | 92.33 | 85.03 | 81.79 |
| | DAPT | 72.20 | 95.82 | 92.27 | 83.03 | 97.06 | 86.55 | 46.37 | 75.99 | 71.38 | 92.65 | 84.53 | 81.62 |
| | PSRC | 73.17 | 96.07 | 93.67 | 83.83 | 97.60 | 87.50 | 50.83 | 77.23 | 72.73 | 92.43 | 86.47 | 82.87 |
| | LLaMP | 73.49 | **97.08** | 94.21 | 86.07 | 98.06 | 87.62 | 56.07 | 77.02 | 74.17 | 91.31 | 86.84 | 83.81 |
| **Diverse** | CoCoOp | 70.83 | 95.16 | 93.34 | 71.57 | 87.84 | 87.25 | 31.21 | 72.15 | 63.04 | 73.32 | 78.14 | 74.90 |
| | PLOT++ | 72.60 | 96.04 | 93.59 | 84.55 | 97.56 | 87.11 | 46.74 | 76.03 | 71.43 | 92.00 | 85.34 | 82.09 |
| | ALIGN | 72.45 | 96.00 | 94.17 | 86.75 | 96.57 | 86.90 | 49.99 | 76.57 | 71.40 | 90.77 | 85.69 | 82.48 |
| | GaLoP | **75.10** | 96.70 | 94.10 | **89.20** | 98.80 | 86.50 | 58.30 | 77.20 | **75.50** | 90.10 | 86.90 | 84.40 |
| | SurPL | 74.65 | 96.92 | **94.22** | 89.00 | **98.84** | **87.63** | **60.51** | **77.67** | 74.75 | **93.92** | **88.18** | **85.12** |

*Table 5.* Base-to-novel generalization performance comparison between our proposed SurPL-G and existing generalizable prompt learning approaches. For fair comparisons, we only list existing inductive methods in the table, without considering the transductive approaches that leverage extra models (e.g., stronger VLMs or LLMs) or data to improve the generalization ability. Averaged results on 11 datasets are reported here. Complete experiment results for individual datasets are given in **appendix**.

| | Base | Novel | HM |
|---|---|---|---|
| CLIP (Radford et al., 2021) | 68.21 | 73.36 | 70.69 |
| CoOp (Zhou et al., 2022c) | 82.68 | 64.15 | 72.25 |
| CoCoOp (Zhou et al., 2022b) | 82.56 | 64.66 | 72.52 |
| ProGrad (Zhu et al., 2023) | 82.48 | 69.12 | 75.21 |
| KgCoOp (Yao et al., 2023) | 81.94 | 72.52 | 76.94 |
| MaPLe (Khattak et al., 2023a) | 82.28 | 75.14 | 78.55 |
| PSRC (Khattak et al., 2023b) | 84.24 | 75.68 | 79.73 |
| ALIGN (Wang et al., 2023) | 83.39 | 75.51 | 79.25 |
| TCP (Yao et al., 2024) | 84.13 | 75.36 | 79.51 |
| DePT (Zhang et al., 2024a) | 85.18 | 76.17 | 80.42 |
| SurPL-G | **86.37** | **76.32** | **81.03** |

diverse text features from different perspectives can significantly benefit the VLM's adaptation ability. Specifically, we observe that the surrogate instance-dependent feature $h^{\text{ID}}$ and fine-grained feature $h^{\text{FG}}$ drastically boost VLM's generalization (base-to-novel results) and discriminative (few-shot results) ability, respectively. This observation manifests that different generated surrogate text features enhance VLM's ability from different perspectives, which also indicates that 1) the generated surrogate text features are sufficiently effective to replace the original diverse prompted text features; 2) our proposed Surrogate Feature Generator is flexible enough to simultaneously generate different diverse prompted text features efficiently.

### 4.3. State-of-the-art Comparison

**Few-shot learning.** As shown in Tab.4, SurPL achieves remarkable few-shot learning performances. Compared

to existing single and diverse prompt learning approaches, SurPL obtains the best average results over 11 datasets. By jointly exploiting different kinds of text features, we surprisingly observe that SurPL even surpasses the state-of-the-art fine-grained-based method GaLoP (Lafon et al., 2024) by 0.72%, while significantly improving the computation efficiency. Beyond the average results, SurPL also achieves the best performances on 7 out of 11 datasets and ranks within the top-2 on all datasets. Notably, for datasets such as FGVCAircraft, EuroSAT and UCF101, SurPL demonstrates substantial improvements over state-of-the-art results, with gains of 2.21%, 1.27% and 1.28%, respectively. The results in Tab.4 clearly show the superiority of SurPL, and verify that the surrogate text features can effectively substitute for original diverse prompted features.

**Base-to-novel Generalization.** Tab.5 compares the generalization ability of SurPL-G with existing generalizable prompt learning methods under the base-to-novel generalization setting. We observe that SurPL-G achieves the highest average results over 11 datasets for both base and novel classes, thereby improving the generalization trade-off results (Harmonic Mean, HM) by 0.61% over the state-of-the-art approach DePT (Zhang et al., 2024a). Specifically, SurPL-G also achieves the best performances on 9 out of 11 datasets. For datasets such as FGVCAircraft, EuroSAT and UCF101, SurPL-G demonstrates substantial improvements over state-of-the-art results, with gains of 1.46%, 2.60% and 1.28%, respectively. The above detailed results are provided in the **appendix**. The results in Tab.5 indicate that by simply incorporating the SAM optimization strategy with SurPL, SurPL-G exhibits compelling generalization ability.

**Cross-dataset Generalization.** We further explore the generalization ability of SurPL-G under the cross-dataset setting, where prompts are learned on ImageNet (source domain) and evaluated on 10 other datasets (target domain). As shown in Tab.6, SurPL-G achieves the best performances on both source and target domains compared to existing

*Table 6.* Cross-dataset generalization performance comparison. Models are learned on ImageNet (source domain) and evaluated on 10 other datasets (target domains). Averaged results of the 10 datasets are reported.

|  | CoOp | CoCoOp | ProGrad | KgCoOp | PLOT | MaPLe | PSRC | DePT | TCP | SurPL-G |
|---|---|---|---|---|---|---|---|---|---|---|
| Source | 71.51 | 71.02 | 72.24 | 70.66 | 71.60 | 70.72 | 71.27 | 71.60 | 71.40 | **73.33** |
| Target (average) | 63.88 | 65.74 | 62.66 | 65.51 | 64.34 | 66.30 | 65.81 | 66.02 | 66.29 | **66.61** |

*Table 7.* Cross-domain generalization performance comparison. V2, Sketch, A and R indicate the ImageNetV2, ImageNet-Sketch, ImageNet-A and ImageNet-R datasets, respectively.

|  | Source | Target | | | | |
|---|---|---|---|---|---|---|
|  | ImageNet | V2 | Sketch | A | R | AVG |
| CoOp | 71.51 | 64.44 | 47.61 | 49.53 | 74.98 | 59.14 |
| CoCoOp | 71.02 | 64.07 | 48.75 | 50.63 | 76.18 | 59.91 |
| ProGrad | 72.24 | 64.27 | 48.10 | 49.72 | 75.84 | 59.48 |
| KgCoOp | 70.66 | 64.10 | 48.97 | 50.69 | 76.70 | 60.12 |
| MaPLe | 70.72 | 64.07 | 49.15 | 50.90 | 76.98 | 60.28 |
| DAPT | 72.20 | 64.93 | 48.30 | 48.74 | 75.75 | 59.43 |
| PSRC | 71.27 | 64.35 | 49.55 | 50.90 | 77.80 | 60.65 |
| TCP | 71.20 | 64.60 | 49.50 | **51.20** | 76.73 | 60.51 |
| SurPL-G | **73.33** | **66.59** | **50.45** | 50.00 | **78.45** | **61.37** |

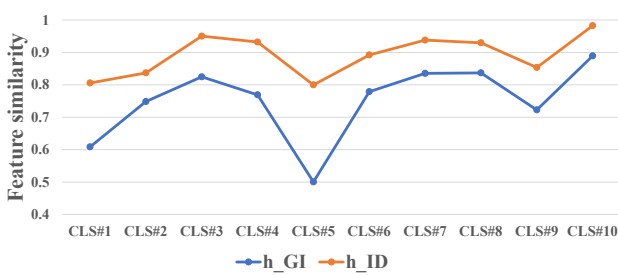

*Figure 4.* Comparison of normalized feature cosine similarity between $\cos(\hat{f}, w^{\text{GI}})$ and $\cos(\hat{f}, h^{\text{ID}})$ across different classes on the UCF101 dataset.

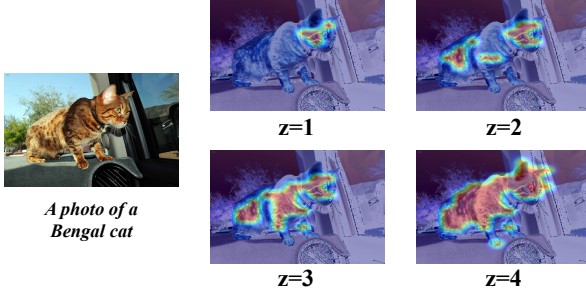

**A photo of a Bengal cat**

*Figure 3.* Visualization results of the attention heat maps of **different surrogate fine-grained text features $h^{\text{FG}} = \{h_z^{\text{FG}}\}_{z=1}^{Z}$.**

methods. This comparison indicates that SurPL-G exhibits a remarkable generalization trade-off.

**Cross-domain Generalization.** We also investigate the generalization ability of SurPL-G under the cross-domain setting, where prompts are learned on ImageNet (source domain) and evaluated on 4 ImageNet-based domain-shifted datasets (target domain). The results illustrated in Tab.7 indicate that SurPL-G reaches significant improvements on both source and target domains over the state-of-the-arts.

### 4.4. In-depth Analysis of Surrogate Features

**Effectiveness of the surrogate fine-grained text features $h^{\text{FG}}$.** As shown in Fig.3, we visualize the attention heat map of different features $\{h_z^{\text{FG}}\}_{z=1}^{Z}$ on a visual concept. The results manifest that 1) our surrogate fine-grained features accurately capture the important information of the visual concept; 2) our surrogate fine-grained features provide a

multi-scale description of the visual concept from coarse to fine. These evidences highlight the high quality of our surrogate text features, further validating the effectiveness of the proposed SurPL.

**Effectiveness of the surrogate instance-dependent text features $h^{\text{ID}}$.** To explore the effectiveness of $h^{\text{ID}}$, we compute the normalized cosine similarity between $h^{\text{ID}}$ and the basic visual feature $\hat{f}$, denoted as $\cos(\hat{f}, h^{\text{ID}})$. Specifically, we randomly sample 10 classes from a dataset and report the average similarity across all visual samples within each class. We also perform the same operation between $\hat{f}$ and the global-invariant text feature $w^{\text{GI}}$: $\cos(\hat{f}, w^{\text{GI}})$. As shown in Fig.4, the comparison results highlight that the surrogate instance-dependent text feature $h^{\text{ID}}$ provides significantly better alignments with the visual concepts, indicating that $h^{\text{ID}}$ effectively captures instance-level knowledge via the proposed Surrogate Feature Generator.

## 5. Conclusion

This paper proposes a novel Surrogate Prompt Learning (SurPL) framework to enable efficient and diverse prompt learning for vision-language models. Instead of learning diverse text prompts from scratch, SurPL directly generates their prompted text features via a lightweight Surrogate Feature Generator, achieving remarkable adaptation performances while maintaining efficiency comparable to single-prompt learning methods. By configuring appropriate conditional signals, SurPL is adaptable and integrable to implement different diverse prompt learning ideas based on any single prompt learning approach. We hope this work

can bring some inspiration to the related fields.

**Limitations and future works.** 1) The Surrogate Feature Generator (SFG) introduced in this paper is inherently a cross-attention module, which is relatively outdated and inevitably increases the parameter sizes. We aim to design more performant and efficient variants in the future. 2) This paper mainly incorporates existing diverse prompt learning frameworks to SurPL. We will further explore effective information that can boost vision-language alignment from new perspectives.

# Acknowledgements

NNW was supported in part by the National Natural Science Foundation of China under Grants U22A2096 and 62036007, in part by Scientific and Technological Innovation Teams in Shaanxi Province under grant 2025RS-CXTD-011, in part by the Shaanxi Province Core Technology Research and Development Project under grant 2024QY2-GJHX-11, in part by the Fundamental Research Funds for the Central Universities under GrantQTZX23042. TLL is partially supported by the following Australian Research Council projects: FT220100318, DP220102121, LP220100527, LP220200949, and IC190100031.

# Impact Statement

This paper presents work whose goal is to advance the field of Machine Learning. There are many potential societal consequences of our work, none which we feel must be specifically highlighted here.

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

## A. Detailed Implementations of Surrogate Feature Generator (SFG)

The parameters of Surrogate Feature Generator $\boldsymbol{\theta}_{\text{SFG}}$ include three input projection layers $\boldsymbol{\theta}_{in1}, \boldsymbol{\theta}_{in2}, \boldsymbol{\theta}_{in3}$, one output projection layer $\boldsymbol{\theta}_{out}$ and two fully connected layers $\boldsymbol{\theta}_{fc1}, \boldsymbol{\theta}_{fc2}$. Given the basic prompted text feature $\boldsymbol{w}$ and the conditional signal $\boldsymbol{\alpha}$, the input Query, Key and Value Matrix of the cross attention can be obtained by:

$$\begin{cases} Q = LN(\boldsymbol{w}) \cdot \boldsymbol{\theta}_{in1}, \\ K = LN(\boldsymbol{\alpha}) \cdot \boldsymbol{\theta}_{in2}, \\ V = LN(\boldsymbol{\alpha}) \cdot \boldsymbol{\theta}_{in3}, \end{cases} \tag{12}$$

where $LN$ is the layer norm operation. Then, the cross attention result $Attn$ can be derived as:

$$Attn = \boldsymbol{w} + \left( softmax\left( \frac{QK^\top}{\sqrt{d_k}} \right) \cdot V \right) \cdot \boldsymbol{\theta}_{out}. \tag{13}$$

$\boldsymbol{\theta}_{in1}, \boldsymbol{\theta}_{in2}, \boldsymbol{\theta}_{in3}, \boldsymbol{\theta}_{out} \in \mathbb{R}^{d \times d}$, where $d$ denotes the feature dimension. Finally, the surrogate feature $\boldsymbol{h}$ is obtained by passing the attention results through a two-layer MLP:

$$\boldsymbol{h} = Attn + ReLU\left( LN(Attn) \cdot \boldsymbol{\theta}_{fc1} \right) \cdot \boldsymbol{\theta}_{fc2}, \tag{14}$$

where $\boldsymbol{\theta}_{fc1} \in \mathbb{R}^{d \times d_{mid}}$ and $\boldsymbol{\theta}_{fc2} \in \mathbb{R}^{d_{mid} \times d}$. To keep SFG parameter-efficient, we set $d_{mid} = d/2$.

Notably, although our proposed SurPL significantly boosts the computation-efficiency, the SFG module inevitably involves extra learnable parameters. Therefore, apart from the mainly concerned computation-efficiency problem, we also discuss the parameter-efficiency of SurPL here. Tab.8 illustrates the parameter size comparison between SurPL and existing methods. We find the total parameter size of SurPL remains acceptable. In fact, it is even smaller than some other prompt learning approaches e.g., MaPLe (Khattak et al., 2023a) and ALIGN (Wang et al., 2023).

*Table 8.* Parameter size comparison between SurPL and existing methods (CoCoOp(Zhou et al., 2022b), PLOT++(Chen et al., 2023), ALIGN(Wang et al., 2023), MaPLe(Khattak et al., 2023a), PSRC(Khattak et al., 2023b), GalLoP(Lafon et al., 2024)).

| Method | CoCoOp | PLOT++ | ALIGN | MaPLe | PSRC | GalLoP | DVLP (baseline) | SurPL |
|--------|--------|--------|-------|-------|------|--------|-----------------|-------|
| Params | 0.035M | 0.014M | 3.582M | 3.556M | 0.046M | 0.606M | 0.061M | 1.641M |

## B. Computation Complexity Analysis of SurPL

Here, we provide a formal analysis of computational complexity of our proposed method SurPL.

**Notations.** loss $\mathcal{L}$, classnames $c = \{c_m\}_{m=1}^{M}$, Text encoder parameter: $T = \{T^k\}_{k=1}^{K}$, SFG parameter: $\boldsymbol{\theta}_{\text{SFG}}$, Text prompts parameter: $s = \{s^k\}_{k=1}^{K}$, Encoder layer depth: $K$, the parameter size of SFG is much smaller than that of text encoder: $\boldsymbol{\theta}_{\text{SFG}} << T$.

**Derivation.** Optimizing the text prompt $s^1$ at the text encoder input position requires computing the gradient through the entire model. For each single surrogate output text feature, the back-propagation computation of SurPL can be written as:

$$\frac{\partial \mathcal{L}}{\partial s^1} = \underbrace{\frac{\partial \mathcal{L}}{\partial \boldsymbol{\theta}_{\text{SFG}}(T^K([s^K, c_m^k]); \alpha_j)} \cdot \frac{\partial \boldsymbol{\theta}_{\text{SFG}}(T^K([s^K, c_m^k]); \alpha_j)}{\partial T^K([s^K, c_m^K])}}_{\text{SFG}} \cdot \underbrace{\frac{\partial T^K([s^K, c_m^K])}{\partial T^{K-1}([s^{K-1}, c_m^{K-1}])} \cdots \frac{\partial T^1([s^1, c_m^1])}{\partial s^1}}_{\text{Text Encoder}}. \tag{15}$$

This gradient computation can be conceptually divided into two parts, which correspond to the text encoder and SFG, respectively. We denote the computational complexity for each output text feature as $\mathcal{O}_T$ and $\mathcal{O}_{\text{SFG}}$.

As shown in Fig.1 (c), SurPL simultaneously involves $M$ text features back-propagation w.r.t text encoder, which equals to the complexity in single prompt learning: $\mathcal{O}(M) = M \cdot \mathcal{O}_T$. For SFG, $(B + Z)M$ text features are involved in computation. The overall complexity of SurPL can thus be expressed as:

$$\mathcal{O}_{SurPL} = M \cdot \mathcal{O}_T + (B + Z)M \cdot \mathcal{O}_{\text{SFG}} \tag{16}$$

Since $\boldsymbol{\theta}_{\text{SFG}} << T$, it follows that $\mathcal{O}_{\text{SFG}} << \mathcal{O}_T$. Therefore, the second term is negligible, and we can approximate: $\mathcal{O}_{SurPL} \approx M \cdot \mathcal{O}_T = \mathcal{O}(M)$.

*Table 9.* Performance comparison between SurPL and SurPL-G under the base-to-novel generalization experiment setting.

| | AVG | | | ImageNet | | | Caltech101 | | |
| --- | --- | --- | --- | --- | --- | --- | --- | --- | --- |
| | Base | Novel | HM | Base | Novel | HM | Base | Novel | HM |
| SurPL | 86.35 | 71.77 | 78.39 | 78.78 | 68.25 | 73.14 | 98.84 | 94.76 | 96.76 |
| SurPL-G | 86.37 | 76.32 | 81.03 | 78.74 | 70.49 | 74.39 | 98.77 | 95.16 | 96.93 |

| | OxfordPets | | | StanfordCars | | | Flowers102 | | |
| --- | --- | --- | --- | --- | --- | --- | --- | --- | --- |
| | Base | Novel | HM | Base | Novel | HM | Base | Novel | HM |
| SurPL | 95.96 | 96.92 | 96.44 | 85.05 | 68.95 | 76.16 | 98.77 | 70.19 | 82.06 |
| SurPL-G | 96.37 | 97.41 | 96.89 | 83.57 | 72.77 | 77.80 | 98.90 | 72.88 | 83.92 |

| | Food101 | | | FGVCAircraft | | | SUN397 | | |
| --- | --- | --- | --- | --- | --- | --- | --- | --- | --- |
| | Base | Novel | HM | Base | Novel | HM | Base | Novel | HM |
| SurPL | 90.27 | 89.97 | 90.12 | 49.52 | 12.96 | 20.54 | 83.19 | 76.55 | 79.73 |
| SurPL-G | 90.92 | 91.81 | 91.36 | 49.20 | 36.93 | 42.19 | 83.43 | 78.96 | 81.13 |

| | DTD | | | EuroSAT | | | UCF101 | | |
| --- | --- | --- | --- | --- | --- | --- | --- | --- | --- |
| | Base | Novel | HM | Base | Novel | HM | Base | Novel | HM |
| SurPL | 85.26 | 55.84 | 67.48 | 95.19 | 77.28 | 85.31 | 89.06 | 77.84 | 83.07 |
| SurPL-G | 86.07 | 62.04 | 72.11 | 94.63 | 81.33 | 87.48 | 89.44 | 79.74 | 84.31 |

## C. SurPL-G: Generalizable version of SurPL by Sharpness-aware Minimization

Recent prompt learning works start to explore the generalization ability of the learned prompts. However, we observe that SurPL exhibits relatively poor generalization performances on unseen tasks. SurPL learns diverse text features to describe visual concepts from different perspectives. While these learned features, especially the surrogate fine-grained features, significantly improve the VLM's discrimination ability, they are also prone to overfitting when trained with Empirical Risk Minimization (ERM), i.e., cross-entropy loss.

To address the above issue, we further propose a generalizable version of SurPL by exploiting sharpness-aware minimization (SAM) optimization strategy (Foret et al., 2021), denoted as SurPL-G. Given the theoretical background that flatter loss landscape yields better model generalization ability, SurPL-G further penalizes the loss sharpness by forcing the neighboring parameters of $\phi$: $(\phi + \epsilon)$ to have the uniformly low loss values, and the corresponding SAM-based optimization loss can be interpreted by:

$$\min_{\phi} \mathcal{L}_{\phi}^{SAM}, \text{ where } \mathcal{L}_{\phi}^{SAM} \triangleq \max_{||\epsilon||_p \leq \rho} \mathcal{L}_{\phi+\epsilon}, \tag{17}$$

where the neighborhood region is controlled by the perturbation radius $\rho$. Eq.17 can be explicitly solved by approximately finding the neighboring parameter $(\phi + \hat{\epsilon})$ that has the maximum loss value using first-order Taylor expansion:

$$\hat{\epsilon} = \underset{||\epsilon||_p \leq \rho}{\text{argmax}} \mathcal{L}_{\phi+\epsilon} \approx \underset{||\epsilon||_p \leq \rho}{\text{argmax}} \mathcal{L}_{\phi} + \epsilon^{\top} \nabla \mathcal{L}_{\phi} = \rho \frac{\nabla \mathcal{L}_{\phi}}{\|\nabla \mathcal{L}_{\phi}\|}. \tag{18}$$

Taking $\hat{\epsilon}$ back to Eq.17, we finally obtain the optimization loss for SurPL-G as shown in Eq.11.

We conduct the comparison experiment between SurPL (optimized via ERM) and SurPL-G (optimized via SAM) under the base-to-novel generalization setting. The results demonstrated in Tab.9 indicate that utilizing the SAM optimization strategy can significantly improve the performances on novel classes. This implementation directly endows SurPL with generalization capabilities without requiring any other modifications.

## D. Algorithms of SurPL and SurPL-G

We provide the pseudo-code of SurPL and SurPL-G in Algorithm.1, which demonstrates the comprehensive and detailed optimization and inference procedures.

## E. Implementation Details of Experiments

**Datasets.** This paper adopts 15 public available visual classification datasets as downstream tasks, including ImageNet (Deng et al., 2009), Caltech101 (Fei-Fei et al., 2004), OxfordPets (Parkhi et al., 2012), StanfordCars (Krause et al., 2013), Flowers102 (Nilsback & Zisserman, 2008), Food101 (Bossard et al., 2014), FGVCAircraft (Maji et al., 2013), EuroSAT

**Algorithm 1** The optimization and inference procedures of SurPL and SurPL-G.

**Input:** Pre-trained VLM with visual-encoder $\mathcal{V}$ and text encoder $\mathcal{T}$, training visual sample: $(\boldsymbol{x}_{train}, y)$, testing visual example $\boldsymbol{x}_{test}$, class names $\boldsymbol{c} = \{\boldsymbol{c}_m\}_{m=1}^M$, text prompt $\boldsymbol{s}$, visual prompts $\boldsymbol{u}$, Surrogate Feature Generator $\boldsymbol{\theta}_{\text{SFG}}$, fine-grained conditional signals $\boldsymbol{\alpha}^{\text{FG}}$, linear projection layer $\boldsymbol{\theta}_{\text{Proj}}$, training epochs $\mathcal{E}$, perturbation radius $\rho$.

1   Initialize the learnable parameters: $\boldsymbol{\phi} = \{\boldsymbol{s}, \boldsymbol{u}, \boldsymbol{\theta}_{\text{SFG}}, \boldsymbol{\alpha}^{\text{FG}}, \boldsymbol{\theta}_{\text{Proj}}\}$
2   **while** *Optimization phase* **do**
3     **for** $i = 1, 2, , ..., \mathcal{E}$ **do**
4        Obtain the basic prompted text feature for each class: $\boldsymbol{w}_m = \mathcal{T}([\boldsymbol{s}, \boldsymbol{c}_m])$
5        Obtain basic and fine-grained visual features: $\hat{\boldsymbol{f}}, \boldsymbol{f} = \mathcal{V}([\boldsymbol{x}_{train}, \boldsymbol{u}]; \boldsymbol{\theta}_{\text{Proj}})$
6        Obtain the global-invariant text feature: $\boldsymbol{w}_m^{GI} = \boldsymbol{w}_m$
7        Obtain the instance-dependent conditional signal $\boldsymbol{\alpha}^{\text{ID}} = \hat{\boldsymbol{f}}$
8        Generate the surrogate instance-dependent text feature $\boldsymbol{h}^{\text{ID}}$: $\boldsymbol{h}_b^{\text{ID}} = \boldsymbol{\theta}_{\text{SFG}}(\boldsymbol{w}, \boldsymbol{\alpha}_b^{\text{ID}})$
9        Generate the surrogate fine-grained text features $\boldsymbol{h}^{\text{FG}}$: $\boldsymbol{h}^{\text{FG}} = \boldsymbol{\theta}_{\text{SFG}}(\boldsymbol{w}, \boldsymbol{\alpha}^{\text{FG}})$
10       Calculate $\mathcal{L}_{CE}^{\text{GI}}$, $\mathcal{L}_{CE}^{\text{ID}}$ and $\mathcal{L}_{CE}^{\text{FG}}$ via Eq.8, Eq.5 and Eq.7, respectively
11       **if** *SurPL* **then**
12         Optimize $\boldsymbol{\phi} = \{\boldsymbol{s}, \boldsymbol{u}, \boldsymbol{\theta}_{\text{SFG}}, \boldsymbol{\alpha}^{\text{FG}}, \boldsymbol{\theta}_{\text{Proj}}\}$ via Eq.9
13       **if** *SurPL-G* **then**
14         Optimize $\boldsymbol{\phi} = \{\boldsymbol{s}, \boldsymbol{u}, \boldsymbol{\theta}_{\text{SFG}}, \boldsymbol{\alpha}^{\text{FG}}, \boldsymbol{\theta}_{\text{Proj}}\}$ via Eq.11
15     **end**
     **Output:** The optimized parameters of $\boldsymbol{\phi} = \{\boldsymbol{s}, \boldsymbol{u}, \boldsymbol{\theta}_{\text{SFG}}, \boldsymbol{\alpha}^{\text{FG}}, \boldsymbol{\theta}_{\text{Proj}}\}$
16   **end**
17   **while** *Inference phase* **do**
18     Repeat step $4 - 9$ on $\boldsymbol{x}_{test}$
19     Calculate $p^{\text{GI}}(m|\boldsymbol{x_{test}})$, $p^{\text{ID}}(m|\boldsymbol{x_{test}})$ and $p_z^{\text{FG}}(m|\boldsymbol{x_{test}})$ via Eq.1, Eq.5 and Eq.7, respectively
20     Calculate the classification probability $p(m|\boldsymbol{x}_{test})$ via Eq.10
     **Output:** The classification probability $p(m|\boldsymbol{x}_{test})$.
21   **end**

(Helber et al., 2019), SUN397 (Xiao et al., 2010), DTD (Cimpoi et al., 2014), UCF101 (Soomro et al., 2012), ImageNetV2 (Recht et al., 2019), ImageNet-Sketch (Wang et al., 2019), ImageNet-A (Hendrycks et al., 2021b) and ImageNet-R (Hendrycks et al., 2021a). These datasets constitute a comprehensive benchmark, which includes classifications of generic objects, scenes, actions, satellites, textures and fine-grained categories. Detailed statistics of the datasets are given in Tab.10.

**Experimental Settings.** We evaluate the effectiveness of the proposed prompt learning method under four experimental settings:

- Few-shot learning: we optimize the parameters of SurPL on few-shot training examples of the downstream task, and evaluate the optimized SurPL by classifying testing data that stem from the same classes.

- Base-to-novel generalization: we split each downstream task into disjoint base and novel classes. The parameters of SurPL are optimized with few-shot training samples of base classes, and evaluated on testing data from novel classes.

- Cross-domain generalization: we optimize the parameters of SurPL on few-shot training examples of ImageNet, and evaluate the optimized SurPL on four domain-shifted datasets: ImageNetV2, ImageNet-Sketch, ImageNet-A and ImageNet-R. Notably, these datasets share the same classnames.

- Cross-dataset generalization: we optimize the parameters of SurPL on few-shot training examples of ImageNet, and evaluate the optimized SurPL on other ten datasets: Caltech101, OxfordPets, StanfordCars, Flowers102, Food101, FGVCAircraft, SUN397, DTD, EuroSAT, UCF101.

**Implementations.** This paper adopts CLIP-ViT-B/16 as the pre-trained VLM for research. DVLP is utilized as the baseline model. The depth and token-length of both text prompts $\boldsymbol{s}$ and visual prompts $\boldsymbol{u}$ are set to $K = 12$ and $L = 4$ by default. All the prompt tokens are initialized by sampling from a zero-mean Gaussian distribution, except for the first-layer text

Table 10. Detailed statistics of the evaluated datasets.

| Dataset | Task | Classes | Training Size | Testing Size |
|---|---|---|---|---|
| ImageNet | Object recognition | 1000 | 1.28M | 50000 |
| Caltech101 | Object recognition | 100 | 4128 | 2465 |
| OxfordPets | Fine-grained pets recognition | 37 | 2944 | 3669 |
| StanfordCars | Fine-grained car recognition | 196 | 6509 | 8041 |
| Flowers102 | Fine-grained flowers recognition | 102 | 4093 | 2463 |
| Food101 | Fine-grained food recognition | 101 | 50500 | 30300 |
| FGVCAircraft | Fine-grained aircraft recognition | 100 | 3334 | 3333 |
| SUN397 | Scene recognition | 397 | 15880 | 19850 |
| DTD | Texture recognition | 47 | 2820 | 1692 |
| EuroSAT | Satellite image recognition | 10 | 13500 | 8100 |
| UCF101 | Action recognition | 101 | 7639 | 3783 |
| ImageNet-V2 | Robustness of collocation | 1000 | N/A | 10000 |
| ImageNet-Sketch | Robustness of sketch domain | 1000 | N/A | 50889 |
| ImageNet-A | Robustness of adversarial attack | 200 | N/A | 7500 |
| ImageNet-R | Robustness of multi-domains | 200 | N/A | 30000 |

Table 11. Hyperparameter settings for few-shot learning and base-to-novel generalization experiments under different datasets.

| | | ImageNet | Caltech101 | OxfordPets | StanfordCars | Flowers102 | Food101 | FGVCAircraft | SUN397 | DTD | EuroSAT | UCF101 |
|---|---|---|---|---|---|---|---|---|---|---|---|---|
| Few-shot | Epochs | 20 | 50 | 50 | 200 | 200 | 10 | 150 | 20 | 50 | 150 | 50 |
| Base-to-novel | Epochs | 10 | 30 | 10 | 30 | 20 | 10 | 30 | 10 | 10 | 10 | 30 |
| | $\rho$ | 0.1 | 0.1 | 0.1 | 0.1 | 0.1 | 0.1 | 0.05 | 0.1 | 0.05 | 0.02 | 0.1 |
| | $K$ | 12 | 12 | 12 | 12 | 12 | 12 | 12 | 12 | 12 | 3 | 12 |

prompt, which is initialized as 'a photo of a'. Similar to GalLoP (Lafon et al., 2024), the number of fine-grained text features and multi-scale constant factor are set as $Z = 4$ and $\eta = 10$. Similar to PSRC (Khattak et al., 2023b), we set the coefficient $\lambda_1 = 25$ and $\lambda_2 = 10$. The perturbation radius in SurPL-G is set to $\rho = 0.1$ by default. Both SurPL and SurPL-G are trained with the SGD optimizer. We initialize the learning rate as $0.0025$ with the cosine annealing decay. A warm-up strategy is applied during the first training epoch, where the learning rate is fixed at $0.00001$. All the experiments are conducted on a single RTX3090 GPU, and results are reported under 16-shot training unless specified. Each result is averaged over three random seeds for a fair comparison.

For the few-shot learning experiment, we set the batch size to $B = 32$ and the number of training epochs to 50 by default. However, we further find that large-scale datasets (e.g., ImageNet and SUN397) benefit from fewer epochs, while datasets containing specific knowledge (e.g., FGVCAircraft, StanfordCars and EuroSAT) perform better with more training epochs. For the base-to-novel generalization experiment, we set the batch size to $B = 4$ and slightly adjust the perturbation radius $\rho$ for very individual datasets. For cross-dataset and cross-domain generalization experiments, we use a batch size of $B = 32$ and set the perturbation radius to $\rho = 0.2$. A summary of hyperparameter adjustments is provided in Tab.11.

## F. Adaptability of SurPL on Different Single Prompt Learning Approaches

As mentioned before, SurPL is adaptable and integrable to implement different diverse prompt learning ideas based on any single prompt learning approach. To this end, we further explore the adaptability of SurPL by implemented our Surrogate Prompt Learning (SurPL) framework on different single prompt learning approaches.

Specifically, we apply CoOp (Zhou et al., 2022c), MaPLe (Khattak et al., 2023a) and PSRC (Khattak et al., 2023b) as baseline models, and incorporates our surrogate instance-dependent and fine-grained text features into each approach. **We strictly follow their implementation details.** The comparison results are presented in Tab.12. By introducing the idea of SurPL into these approaches, we observe significant performance improvements for CoOp, MaPLe and PSRC, achieving average gains of $3.10\%$, $2.36\%$ and $1.24\%$, respectively. This comparison further demonstrates the effectiveness and adaptability of our proposed SurPL.

*Table 12.* Performance comparison between w/o and w/ SurPL on different single prompt learning approaches.

|  | ImageNet | Caltech101 | Pets | Cars | Flowers | Food101 | Aircraft | Sun397 | DTD | EuroSAT | UCF101 | AVG |
|---|---|---|---|---|---|---|---|---|---|---|---|---|
| CoOp | 71.87 | 95.57 | 91.87 | 83.07 | 97.07 | 84.20 | 43.40 | 74.67 | 69.87 | 84.93 | 82.23 | 79.89 |
| +SurPL | 73.87 | 96.56 | 93.63 | 87.03 | 98.24 | 86.29 | 52.41 | 77.10 | 73.88 | 87.96 | 85.90 | 82.99 |
| Δ | +2.00 | +0.99 | +1.76 | +3.96 | +1.17 | +2.09 | +9.01 | +2.43 | +4.01 | +3.03 | +3.67 | +3.10 |
| MaPLe | 72.33 | 96.00 | 92.83 | 83.57 | 97.00 | 85.33 | 48.40 | 75.53 | 71.33 | 92.33 | 85.03 | 81.79 |
| +SurPL | 73.98 | 96.67 | 93.71 | 87.79 | 98.37 | 86.38 | 55.76 | 77.45 | 74.33 | 93.49 | 87.76 | 84.15 |
| Δ | +1.65 | +0.67 | +0.88 | +4.22 | +1.37 | +1.05 | +7.36 | +1.92 | +3.00 | +1.16 | +2.73 | +2.36 |
| PSRC | 73.17 | 96.07 | 93.67 | 83.83 | 97.60 | 87.50 | 50.83 | 77.23 | 72.73 | 92.43 | 86.47 | 82.87 |
| +SurPL | 74.95 | 96.96 | 94.25 | 86.99 | 98.27 | 87.42 | 53.04 | 78.22 | 75.02 | 92.37 | 87.66 | 84.10 |
| Δ | +1.78 | +0.89 | +0.58 | +3.16 | +0.67 | -0.08 | +2.21 | +0.99 | +2.29 | -0.06 | +1.19 | +1.24 |

*Table 13.* Performance comparison with other PEFT methods.

| Method | ImageNet | Caltech101 | Pets | Cars | Flowers | Food101 | Aircraft | Sun397 | DTD | EuroSAT | UCF101 | AVG |
|---|---|---|---|---|---|---|---|---|---|---|---|---|
| CLIP-Adapter | 71.60 | 94.57 | 92.03 | 80.90 | 97.00 | 86.83 | 42.67 | 75.30 | 71.17 | 81.87 | 84.53 | 79.86 |
| Tip-Adapter | 73.10 | 95.79 | 92.70 | 83.09 | 96.18 | 87.24 | 45.59 | 74.99 | 72.05 | 87.46 | 84.50 | 81.15 |
| TaskRes | 73.00 | 95.80 | 92.40 | 83.50 | 97.50 | 86.90 | 44.90 | 76.10 | 71.50 | 82.70 | 84.00 | 80.75 |
| CLIPFit | 71.53 | 96.13 | 93.50 | 82.43 | 96.37 | 87.37 | 45.47 | 75.67 | 71.57 | 90.13 | 83.83 | 81.27 |
| CLIP-LoRA | 73.60 | 96.40 | 92.40 | 86.30 | 98.00 | 84.20 | 54.70 | 76.10 | 72.00 | 92.10 | 86.70 | 82.95 |
| SurPL | 74.65 | 96.92 | 94.22 | 89.00 | 98.84 | 87.63 | 60.51 | 77.67 | 74.75 | 93.92 | 88.18 | 85.12 |

## G. Comparison with Other PEFT Methods

Apart from prompt learning, we also compare SurPL with other parameter-efficient fine-tuning (PEFT) methods including adapter (CLIP-adapter (Gao et al., 2024), Tip-Adapter (Zhang et al., 2022) and TaskRes (Yu et al., 2023)), BitFit (CLIPFit (Li et al., 2024)) and LoRA (CLIP-LoRA (Zanella & Ben Ayed, 2024)) based approaches in Tab.13. The results indicate that SurPL also outperforms them by a significant margin.

Adapter-based methods utilize lightweight networks to adapt the text and visual features output by the VLM's encoders. While our proposed Surrogate Feature Generator (SFG) and adapter both adapt the output text features, we clarify the different purposes and motivations between them. SFG aims to generate **novel** text features based on the output text feature, which enables the description of visual concepts from different perspectives. Adapter mainly targets to simply self-enhance the discriminative ability of the output text feature.

## H. Additional Experiment Results

- Performance comparison under different number of few-shot learning (Tab.14).

- Complete ablation study results of each individual datasets under few-shot learning setting (Tab.15).

- Complete ablation study results of each individual datasets under base-to-novel generalization setting (Tab.16).

- Complete experiment results of each individual datasets under base-to-novel generalization setting (Tab.17).

- Complete experiment results of each individual datasets under cross-dataset generalization setting (Tab.18).

- Additional visualization results of the heat maps (Fig.5).

Table 14. Performance comparison under different number of few-shot learning.

|  | 2-shot | 4-shot | 8-shot | 16-shot |
|---|---|---|---|---|
| CoOp (Zhou et al., 2022c) | 70.65 | 74.02 | 76.98 | 79.89 |
| CoCoOp (Zhou et al., 2022b) | 67.65 | 71.21 | 72.96 | 74.90 |
| MaPLe (Khattak et al., 2023a) | 72.58 | 72.58 | 78.89 | 81.79 |
| PLOT (Chen et al., 2023) | 74.00 | 76.90 | 79.60 | 82.10 |
| PSRC (Khattak et al., 2023b) | 75.29 | 78.35 | 80.69 | 82.87 |
| ALIGN (Wang et al., 2023) | 73.56 | 76.85 | 79.58 | 82.48 |
| GalLoP (Lafon et al., 2024) | **76.40** | 79.10 | 82.20 | 84.50 |
| SurPL | 75.39 | **79.24** | **82.52** | **85.12** |

Table 15. Complete ablation study results of each individual datasets under few-shot learning setting.

|  | ImageNet | Caltech101 | Pets | Cars | Flowers | Food101 | Aircraft | Sun397 | DTD | EuroSAT | UCF101 | AVG |
|---|---|---|---|---|---|---|---|---|---|---|---|---|
| DVLP | 72.62 | 96.24 | 93.14 | 85.75 | 97.84 | 87.30 | 52.88 | 76.15 | 71.96 | 92.58 | 85.64 | 82.92 |
| GI | 73.27 | 96.16 | 93.90 | 86.67 | 98.10 | 87.34 | 55.25 | 76.88 | 73.37 | 93.01 | 86.70 | 83.70 |
| GI+ID | 73.46 | 96.47 | 93.25 | 86.88 | 98.34 | 87.43 | 55.46 | 77.07 | 74.23 | 93.67 | 87.32 | 83.96 |
| GI+FG | 74.47 | 96.77 | 94.22 | 89.46 | 98.78 | 87.29 | 60.88 | 77.29 | 74.64 | 93.85 | 87.20 | 84.99 |
| GI+ID+FG | 74.65 | 96.92 | 94.22 | 89.00 | 98.84 | 87.63 | 60.51 | 77.67 | 74.75 | 93.92 | 88.18 | 85.12 |

Table 16. Complete ablation study results of each individual datasets under base-to-novel generalization setting.

|  | AVG | | | ImageNet | | | Caltech101 | | | OxfordPets | | | StanfordCars | | | Flowers102 | | |
|---|---|---|---|---|---|---|---|---|---|---|---|---|---|---|---|---|---|---|
|  | Base | Novel | HM | Base | Novel | HM | Base | Novel | HM | Base | Novel | HM | Base | Novel | HM | Base | Novel | HM |
| DVLP | 82.30 | 74.25 | 78.06 | 75.98 | 71.44 | 73.64 | 98.15 | 95.09 | 96.60 | 96.12 | 97.93 | 97.02 | 72.81 | 75.05 | 73.91 | 97.66 | 74.14 | 84.29 |
| GI | 81.84 | 74.68 | 78.09 | 75.57 | 71.15 | 73.29 | 97.78 | 94.80 | 96.27 | 95.80 | 97.50 | 96.64 | 69.48 | 76.05 | 72.62 | 97.37 | 77.18 | 86.11 |
| GI+ID | 84.77 | 76.19 | 80.25 | 76.94 | 71.67 | 74.21 | 98.28 | 95.01 | 96.62 | 95.89 | 97.44 | 96.66 | 77.87 | 75.05 | 76.43 | 98.67 | 74.99 | 85.22 |
| GI+FG | 85.52 | 75.50 | 80.20 | 78.28 | 69.09 | 73.40 | 98.92 | 94.65 | 96.74 | 96.21 | 97.03 | 96.62 | 81.62 | 72.20 | 76.62 | 98.86 | 72.89 | 83.91 |
| GI+ID+FG | 86.37 | 76.32 | 81.03 | 78.74 | 70.49 | 74.39 | 98.77 | 95.16 | 96.93 | 96.37 | 97.41 | 96.89 | 83.57 | 72.77 | 77.80 | 98.9 | 72.88 | 83.92 |

|  | Food101 | | | FGVCAircraft | | | SUN397 | | | DTD | | | EuroSAT | | | UCF101 | | |
|---|---|---|---|---|---|---|---|---|---|---|---|---|---|---|---|---|---|---|
|  | Base | Novel | HM | Base | Novel | HM | Base | Novel | HM | Base | Novel | HM | Base | Novel | HM | Base | Novel | HM |
| DVLP | 90.87 | 92.10 | 91.48 | 42.64 | 36.49 | 39.33 | 79.81 | 79.49 | 79.65 | 83.29 | 59.50 | 69.41 | 81.94 | 56.94 | 67.19 | 85.99 | 78.55 | 82.10 |
| GI | 90.61 | 91.97 | 91.28 | 41.08 | 29.63 | 34.43 | 79.00 | 79.76 | 79.38 | 82.91 | 61.64 | 70.71 | 84.94 | 61.26 | 71.18 | 85.68 | 80.51 | 83.01 |
| GI+ID | 90.87 | 92.06 | 91.46 | 47.12 | 37.29 | 41.63 | 81.90 | 79.85 | 80.86 | 84.45 | 61.55 | 71.20 | 91.93 | 71.83 | 80.65 | 88.52 | 81.32 | 84.77 |
| GI+FG | 90.42 | 91.09 | 90.75 | 45.06 | 36.43 | 40.29 | 82.82 | 77.76 | 80.21 | 86.46 | 62.60 | 72.62 | 93.60 | 77.14 | 84.58 | 88.52 | 79.65 | 83.85 |
| GI+ID+FG | 90.92 | 91.81 | 91.36 | 49.20 | 36.93 | 42.19 | 83.43 | 78.96 | 81.13 | 86.07 | 62.04 | 72.11 | 94.63 | 81.33 | 87.48 | 89.44 | 79.74 | 84.31 |

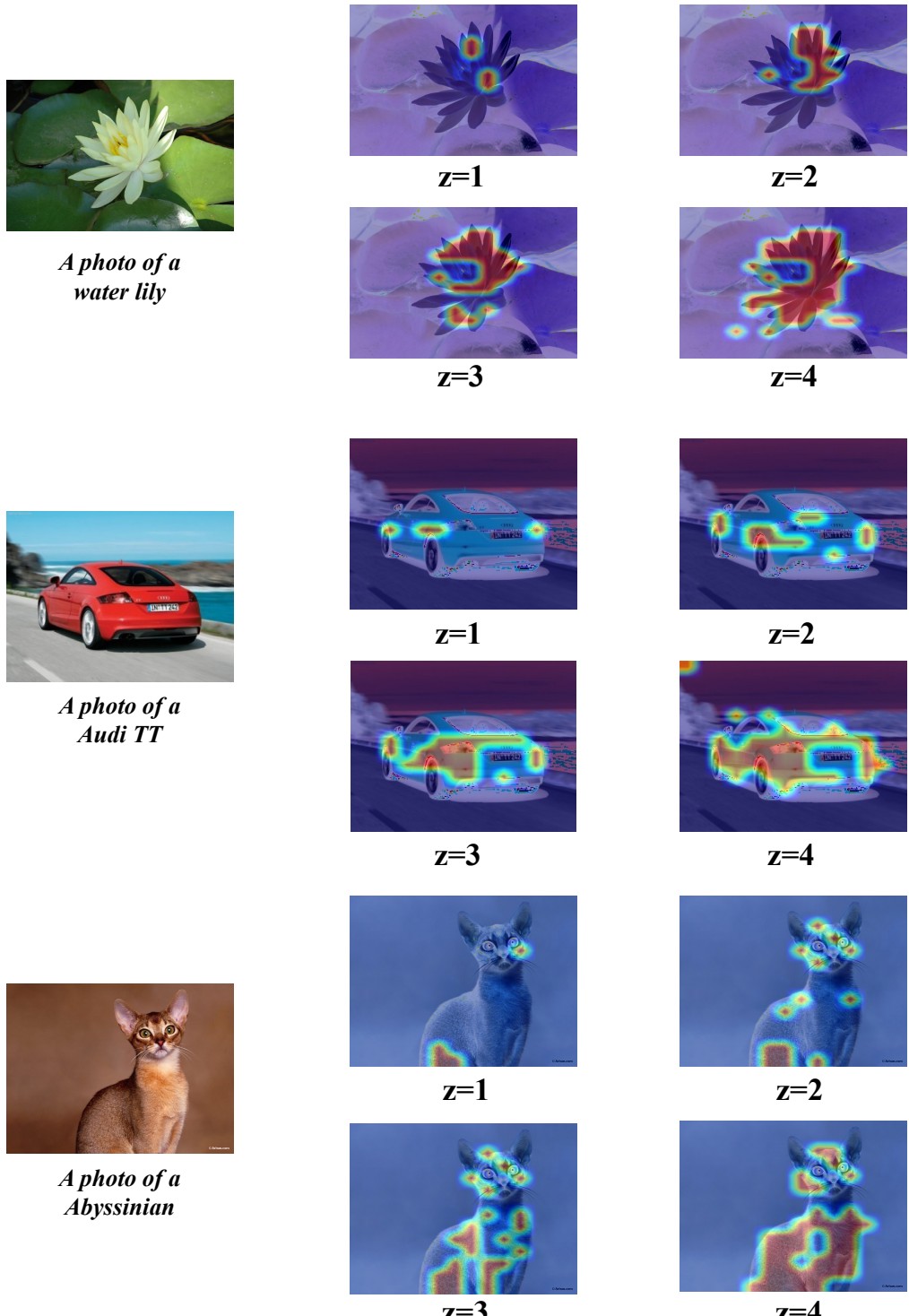

*Figure 5.* Additional visualization results of the attention heat maps.

*Table 17.* Complete experiment results of each individual datasets under base-to-novel generalization setting.

| | AVG | | | ImageNet | | | Caltech101 | | | OxfordPets | | | StanfordCars | | | Flowers102 | | |
| --- | --- | --- | --- | --- | --- | --- | --- | --- | --- | --- | --- | --- | --- | --- | --- | --- | --- | --- |
| | Base | Novel | HM | Base | Novel | HM | Base | Novel | HM | Base | Novel | HM | Base | Novel | HM | Base | Novel | HM |
| CLIP | 68.21 | 73.36 | 70.69 | 72.42 | 68.13 | 70.21 | 97.29 | 94.10 | 95.67 | 89.47 | 96.81 | 93.00 | 63.87 | 74.97 | 68.98 | 69.23 | 76.74 | 72.79 |
| CoOp | 82.68 | 64.15 | 72.25 | 76.34 | 65.17 | 70.31 | 98.15 | 93.23 | 95.63 | 94.19 | 96.01 | 95.09 | 77.58 | 65.02 | 70.75 | 97.79 | 63.95 | 77.33 |
| CoCoOp | 82.56 | 64.66 | 72.52 | 76.09 | 69.73 | 72.77 | 97.92 | 91.78 | 94.75 | 93.48 | 95.77 | 94.61 | 77.65 | 63.00 | 69.56 | 97.44 | 60.97 | 75.01 |
| ProGrad | 82.48 | 69.12 | 75.21 | 76.72 | 67.80 | 71.98 | 98.30 | 93.96 | 96.08 | 94.59 | 96.98 | 95.77 | 77.11 | 69.89 | 73.32 | 96.52 | 69.86 | 81.05 |
| KgCoOp | 81.94 | 72.52 | 76.94 | 75.89 | 69.55 | 72.58 | 97.72 | 94.32 | 95.99 | 95.16 | 96.61 | 95.88 | 74.16 | 74.64 | 74.40 | 95.85 | 73.19 | 83.00 |
| MaPLe | 82.28 | 75.14 | 78.55 | 76.66 | 70.54 | 73.47 | 97.74 | 94.36 | 96.02 | 95.43 | 97.76 | 96.58 | 72.94 | 74.00 | 73.47 | 95.92 | 72.46 | 82.56 |
| PSRC | 84.24 | 75.68 | 79.73 | 77.62 | 70.55 | 73.92 | 98.06 | 93.85 | 95.91 | 95.20 | 97.31 | 96.24 | 78.42 | 75.26 | 76.81 | 98.01 | 77.07 | 86.29 |
| ALIGN | 83.39 | 75.51 | 79.25 | 76.89 | **72.15** | 74.44 | 98.37 | 94.70 | 96.50 | 95.67 | **97.93** | 96.79 | 77.24 | **76.38** | 76.81 | 97.70 | 73.30 | 83.76 |
| TCP | 84.13 | 75.36 | 79.51 | 77.27 | 69.87 | 73.38 | 98.23 | 94.67 | 96.42 | 94.67 | 97.20 | 95.92 | 80.80 | 74.13 | 77.32 | 97.73 | 75.57 | 85.23 |
| DePT | 85.18 | 76.17 | 80.42 | 78.20 | 70.27 | 74.02 | 98.57 | 94.10 | 96.28 | 95.43 | 97.33 | 96.37 | 80.80 | 75.00 | 77.79 | 98.40 | **77.10** | **86.46** |
| SurPL-G | **86.37** | 76.32 | **81.03** | **78.74** | 70.49 | **74.39** | **98.77** | **95.16** | **96.93** | **96.37** | 97.41 | **96.89** | **83.57** | 72.77 | **77.80** | **98.90** | 72.88 | 83.92 |

| | Food101 | | | FGVCAircraft | | | SUN397 | | | DTD | | | EuroSAT | | | UCF101 | | |
| --- | --- | --- | --- | --- | --- | --- | --- | --- | --- | --- | --- | --- | --- | --- | --- | --- | --- | --- |
| | Base | Novel | HM | Base | Novel | HM | Base | Novel | HM | Base | Novel | HM | Base | Novel | HM | Base | Novel | HM |
| CLIP | 89.42 | 90.68 | 90.05 | 27.55 | 33.29 | 30.15 | 69.40 | 75.56 | 72.35 | 53.36 | 51.69 | 52.51 | 50.19 | 69.90 | 58.43 | 68.10 | 75.12 | 71.44 |
| CoOp | 88.68 | 85.31 | 86.96 | 39.94 | 24.62 | 30.46 | 80.58 | 63.90 | 71.28 | 79.82 | 45.17 | 57.69 | 91.25 | 47.26 | 62.27 | 85.20 | 56.05 | 67.62 |
| CoCoOp | 88.11 | 83.92 | 85.96 | 41.80 | 25.08 | 31.35 | 79.38 | 67.99 | 73.24 | 79.94 | 42.55 | 55.54 | 92.14 | 51.33 | 65.93 | 84.25 | 59.17 | 69.52 |
| ProGrad | 90.11 | 89.56 | 89.83 | 40.30 | 25.81 | 31.47 | 81.11 | 71.31 | 75.89 | 76.85 | 51.89 | 61.95 | 91.09 | 56.21 | 69.52 | 84.59 | 67.03 | 74.79 |
| KgCoOp | 90.53 | 91.01 | 90.77 | 38.72 | 29.63 | 33.57 | 80.71 | 76.28 | 78.43 | 80.44 | 56.69 | 66.51 | 88.15 | 60.42 | 71.70 | 84.00 | 75.37 | 79.45 |
| MaPLe | 90.71 | 92.05 | 91.38 | 37.44 | 35.61 | 36.50 | 80.82 | 78.70 | 79.75 | 80.36 | 59.18 | 68.16 | 94.07 | 73.23 | 82.35 | 83.00 | 78.66 | 80.77 |
| PSRC | 90.66 | 91.56 | 91.11 | 42.86 | 36.59 | 39.48 | 82.54 | 78.81 | 80.63 | 83.53 | 59.62 | 69.58 | 92.85 | 72.95 | 81.71 | 86.94 | 78.92 | 82.74 |
| ALIGN | 90.77 | **92.07** | **91.42** | 37.56 | **36.97** | 37.26 | 82.47 | **79.68** | 81.05 | 82.13 | 54.17 | 65.28 | 94.03 | 74.9 | 83.38 | 84.43 | 78.33 | 81.27 |
| TCP | 90.57 | 91.37 | 90.97 | 41.97 | 34.43 | 37.83 | 82.63 | 78.20 | 80.35 | 82.77 | 58.07 | 68.25 | 91.63 | 74.73 | 82.32 | 87.13 | **80.77** | 83.83 |
| DePT | 90.87 | 91.57 | 91.22 | 45.70 | 36.73 | 40.73 | 83.27 | 78.97 | 81.06 | 84.80 | 61.20 | 71.09 | 93.23 | 77.90 | 84.88 | 87.73 | 77.70 | 82.41 |
| SurPL-G | **90.92** | 91.81 | 91.36 | **49.20** | 36.93 | **42.19** | **83.43** | 78.96 | **81.13** | **86.07** | 62.04 | **72.11** | **94.63** | 81.33 | **87.48** | **89.44** | 79.74 | **84.31** |

*Table 18.* Complete experiment results of each individual datasets under cross-dataset generalization setting.

| | ImageNet | Caltech101 | Pets | Cars | Flowers | Food101 | Aircraft | Sun397 | DTD | EuroSAT | UCF101 | AVG |
| --- | --- | --- | --- | --- | --- | --- | --- | --- | --- | --- | --- | --- |
| CLIP | 66.70 | 93.30 | 89.10 | 65.70 | 70.70 | 85.90 | 24.90 | 62.60 | 44.30 | 48.30 | 67.60 | 65.24 |
| CoOp | 71.51 | 93.70 | 89.14 | 64.51 | 68.71 | 85.30 | 18.47 | 64.15 | 41.92 | 46.39 | 66.55 | 63.88 |
| CoCoOp | 71.02 | 94.43 | 90.14 | 65.32 | 71.88 | 86.06 | 22.94 | 67.36 | 45.73 | 45.37 | 68.21 | 65.74 |
| ProGrad | 72.24 | 91.52 | 89.64 | 62.39 | 67.87 | 85.40 | 20.16 | 62.47 | 39.42 | 43.46 | 64.29 | 62.66 |
| KgCoOp | 70.66 | 93.92 | 89.83 | 65.41 | 70.01 | 86.36 | 22.51 | 66.16 | 46.35 | 46.04 | 68.50 | 65.51 |
| PLOT | 71.60 | 92.07 | 90.10 | 65.70 | 69.23 | 86.23 | 25.00 | 61.67 | 38.60 | 47.83 | 67.00 | 64.34 |
| MaPLe | 70.72 | 93.53 | 90.49 | 65.57 | 72.20 | 86.20 | 24.74 | 67.01 | 46.49 | 48.06 | 68.69 | 66.30 |
| PSRC | 71.27 | 93.60 | 90.25 | 65.70 | 70.25 | 86.15 | 23.90 | 67.10 | 46.87 | 45.50 | 68.75 | 65.81 |
| DePT | 71.60 | 93.80 | 90.13 | 66.00 | 70.93 | 86.27 | 24.30 | 67.23 | 46.60 | 45.83 | 69.10 | 66.02 |
| TCP | 71.40 | 93.97 | 91.25 | 64.69 | 71.21 | 86.69 | 23.45 | 67.15 | 44.35 | 51.45 | 68.73 | 66.29 |
| SurPL-G | 73.33 | 93.73 | 90.16 | 64.67 | 70.53 | 85.52 | 24.80 | 67.43 | 48.54 | 52.85 | 67.86 | 66.61 |

