# OpenReview forum: "Surrogate Prompt Learning: Towards Efficient and Diverse Prompt Learning for Vision-Language Models"
_ICML.cc/2025/Conference — ICML 2025 poster_

### Official Review · Reviewer_4GML · 2025-03-02

**Overall Recommendation:** 3

**Summary:**

This paper presents a novel Surrogate Prompt Learning (SurPL) framework for vision-language models (VLMs). SurPL aims to achieve efficient and diverse prompt learning by replacing explicit diverse prompt learning with a Surrogate Feature Generator (SFG) that generates diverse text features without requiring complex gradient computations. This approach significantly reduces the computational burden while maintaining competitive performance. The authors provide extensive experimental validation across multiple benchmarks, showing that SurPL improves both efficiency and performance compared to existing prompt learning methods. The paper also introduces SurPL-G, an extension designed to improve generalization through Sharpness-aware Minimization (SAM).

**Claims And Evidence:**

Efficiency Improvement

Claim: SurPL reduces computational costs by avoiding gradient-based optimization for diverse prompts.
Evidence: Tables 1 and 2 compare SurPL with diverse and single-prompt learning baselines, showing reduced GPU memory usage and faster training/testing times.
Performance Enhancement

Claim: SurPL achieves state-of-the-art accuracy while being computationally efficient.
Evidence: Tables 4 and 5 demonstrate that SurPL outperforms existing diverse and single-prompt methods in few-shot learning and generalization.
Flexibility in Prompt Learning

Claim: SurPL generalizes to multiple diverse prompt learning strategies.
Evidence: The authors show how SurPL can generate instance-dependent and fine-grained prompt representations.
Generalization with SAM (SurPL-G)

Claim: SurPL-G enhances generalization using SAM.
Evidence: Table 5 shows that SurPL-G achieves superior performance on base-to-novel generalization.

**Essential References Not Discussed:**

The paper could benefit from a discussion on meta-learning approaches for efficient adaptation, such as:

- Ha, Hyeonmin, et al. "Meta-Learning of Prompt Generation for Lightweight Prompt Engineering on Language-Model-as-a-Service." Findings of the Association for Computational Linguistics: EMNLP 2023. 2023.

**Experimental Designs Or Analyses:**

Strengths:

The experimental setup is comprehensive and includes efficiency, accuracy, and generalization evaluations.
The comparisons are fair, using the same baseline (Dense Visual-Language Prompt, DVLP).
Ablation studies effectively demonstrate the impact of different components.
Weaknesses:

There is no qualitative analysis of how surrogate text features differ from learned diverse prompts.
The paper does not discuss potential failure cases (e.g., when SurPL might underperform compared to explicit prompt learning).
Hyperparameter sensitivity analysis is missing.

**Methods And Evaluation Criteria:**

Methods and Evaluation Criteria
The paper employs rigorous evaluation methodologies:

Benchmarks: 15 widely used datasets (e.g., ImageNet, EuroSAT, UCF101).
Comparison with state-of-the-art methods: Covers both single and diverse prompt learning approaches.
Ablation studies: Justifies the effectiveness of individual components (e.g., instance-dependent and fine-grained surrogate prompts).
Efficiency analysis: Demonstrates reduced GPU memory and computational overhead.
Potential Issues:

The paper does not explore the impact of different types of conditional signals on surrogate text feature generation.
While the SAM optimization improves generalization, the effect of different perturbation radii is not analyzed in detail.

**Other Comments Or Suggestions:**

Feature Interpretability: Provide visualization of the generated surrogate features.

Robustness Analysis: Evaluate the sensitivity of SurPL to different choices of conditional signals.

Ablation on Surrogate Feature Generator: Analyze how reducing SFG complexity affects performance.

**Other Strengths And Weaknesses:**

Strengths

Addresses an important limitation (high computation cost of diverse prompt learning).

Provides a strong empirical validation across various benchmarks.

The proposed method is flexible and compatible with existing VLMs.

Weaknesses

The theoretical foundations of surrogate feature generation are not well explored.

No qualitative analysis (e.g., feature visualization) to show how surrogate features compare to explicitly learned prompts.

The limitations section is brief and does not discuss potential failure modes.

**Questions For Authors:**

How does SurPL handle domain shifts compared to explicit diverse prompt learning?

Can the surrogate feature generator be trained separately from the main model?

Does SurPL introduce additional latency during inference?

How sensitive is SurPL to the choice of conditional signals?

Have you considered using knowledge distillation techniques to refine surrogate feature generation?

**Relation To Broader Scientific Literature:**

The paper builds on prior work in:

Prompt Learning: CoOp (Zhou et al., 2022), CoCoOp (Zhou et al., 2022b), PSRC (Khattak et al., 2023b).

Vision-Language Models: CLIP (Radford et al., 2021), ALIGN (Jia et al., 2021).

Generalization Techniques: Sharpness-aware Minimization (SAM) (Foret et al., 2021).

The work is well-motivated and aligns with the ongoing trend of efficient fine-tuning methods for large pre-trained models.

**Theoretical Claims:**

The paper does not introduce significant new theoretical contributions but relies on well-established concepts like contrastive learning, attention mechanisms, and SAM.
The derivations of loss functions (Equations 1-11) appear correct but lack deeper theoretical justification on why surrogate features retain meaningful semantic information.

---

> ### Author Rebuttal · Authors · 2025-04-01
>
> >Q1: Lack deeper theoretical justification on why surrogate features retain meaningful semantic information.
>
> A1: Thanks for the comments. We provide a theoretical analysis based on the universal approximation theorem. Due to character limits, please refer to the Q&A 1 of Reviewer jTad for detailed analysis.
>
> >Q2: No qualitative comparison between surrogate and explicitly learned prompts.
>
> A2: Thanks for the insightful comments. We provide direct visualization map comparisons between surrogate and explicit FG prompts (due to the computation resource limit, we can’t afford to reproduce explicit ID prompt learning). As shown in [explicit-versus-surrogate.png](https://postimg.cc/sQGnBFP5), heatmaps generated from explicit and surrogate prompts focus on the same regions of the image, particularly at large FG scales ($Z=3,4$). For small FG scales ($Z=1,2$), while the heatmaps show minor differences, they still consistently focus on the target object. These comparison results indicate the effectiveness of SFG.
>
> >Q3: No discussion about the failure cases of SurPL.
>
> A3: We first provide the direct comparison between explicit FG prompt learning and surrogate FG prompt learning on 9 datasets (except ImageNet and SUN397 due to the computational resource limit). The averaged results indicate surrogate features don’t exhibit significant failures compared to explicit features.
>
> | Explicit | Surrogate |
> |:--------:|:---------:|
> |   86.79  |   86.70   |
>
> We further compare the generalization ability between explicit and surrogate prompts on 10 datasets (excluding ImageNet). The results indicate that surrogate prompts initially exhibit weaker generalization ability. We attribute it to the lightweight architecture of SFG. Generating features from SFG may overfit more easily. However, through our proposed SurPL-G, we effectively address this limitation and achieve remarkable performance.
>
> |               |  Base | Novel |   HM  |
> |:-------------:|:-----:|:-----:|:-----:|
> |    Explicit   | 86.76 | 73.15 | 79.38 |
> |   Surrogate   | 87.11 | 72.13 | 78.91 |
> | Surrogate+SAM | 87.13 | 76.90 | 81.70 |
>
> Finally, we compare SurPL with the state-of-the-art work GalLop. While GalLoP achieves marginally superior results on certain datasets (e.g., ImageNet), we attribute this advantage to its multiple global prompt learning and dropout strategy. This strategy induces diversity through randomization, enhancing the performance. We will further explore how to incorporate such strategy into our SurPL.
>
> >Q4: Hyperparameter sensitivity analysis is missing.
>
> A4: We provide hyperparameter analysis at the Q&A 2 of Reviewer jTad. The results indicate that the hyperparameters applied are reasonable and relatively stable.
>
> >Q5: Analyze how reducing SFG complexity affects performance.
>
> A5: We appreciate this insightful suggestion. Due to the time limit, we only explore this trade-off by reducing the dimension of $d_{mid}$ in $\theta _ {fc1}$ and $\theta _ {fc2}$ in SFG. The results indicate that accuracy remains relatively stable despite parameter reduction. This inspires us to further reduce the parameter of projection layers $\theta _ {in1}$, $\theta _ {in2}$, $\theta _ {in3}$ and $\theta _ {out}$.
>
> | $d_{mid}$ | Param(M) |  Acc  |
> |:-------:|:--------:|:-----:|
> |   256   |   1.31   | 85.12 |
> |   128   |   1.18   | 85.00 |
> |    64   |   1.11   | 84.99 |
> |    32   |   1.08   | 85.00 |
>
> >Q6: How does SurPL handle domain shifts compared to explicit diverse prompt learning?
>
> A6: Thanks for the question. Domain shift tightly associates with model’s generalization ability. Due to the computation resource limit, we can’t reproduce explicit prompt learning on cross-domain experiment (based on ImageNet). We alternatively conduct the comparison of explicit and surrogate prompt learning on base-to-novel setting. Please refer to Q&A 3 for detailed analysis.
>
> >Q7: How sensitive is SurPL to the choice of conditional signals?
>
> A7: Thanks for the question. The conditional signals are pre-defined according to the existing diverse prompt learning methods. Please refer to Q&A 4 of Reviewer Zno2 for details.
>
> >Q8: Can SFG be trained separately from the main model? Consider knowledge distillation techniques to refine surrogate feature generation.
>
> A8: Thanks for the valuable suggestion. At this stage, SFG cannot be trained separately since it is intrinsically a fine-tuning approach and builds upon $w$. We will keep on research to explore the idea of applying pre-trained models and knowledge distillation to replace the cross-attention module for SFG.
>
> >Q9: Does SurPL introduce additional latency during inference?
>
> A9: No significant additions compared with baseline DVLP. Please refer to Table.2 for details.
>
> >Q10: Effect of $\rho$.
>
> A10: We explore the effect of $\rho$ on cross-dataset setting. The results indicate a seen-unseen trade-off.
>
> |     |  Seen | Unseen |
> |:---:|:-----:|:------:|
> | 0.1 |  74.2 |  65.13 |
> | 0.2 | 73.33 |  66.61 |
> | 0.3 | 72.74 |  66.74 |

---

### Official Review · Reviewer_jTad · 2025-03-08

**Overall Recommendation:** 3

**Summary:**

This paper introduces Surrogate Prompt Learning to address efficiency issues in diverse prompt learning for vision-language models (VLMs). SurPL leverages a lightweight Surrogate Feature Generator (SFG) to directly generate diverse prompted text features from a single basic prompt, avoiding the computational overhead of conventional approaches. Experiments across 15 datasets show SurPL's effectiveness.

## update after rebuttal
The author has adequately addressed my concerns, thanks!

**Claims And Evidence:**

The paper provides empirical validation through experiments on 15 different datasets across multiple settings. The efficiency claims are backed by direct comparisons of GPU memory usage, training time, and testing time. The effectiveness of the Surrogate Feature Generator is demonstrated through ablation studies, and the visualizations in Figures 3 and 4 provide qualitative evidence of the surrogate features' ability to capture relevant visual information.

**Essential References Not Discussed:**

Nil.

**Experimental Designs Or Analyses:**

Some potential improvements to the experimental analysis could include:

1. The paper doesn't provide much theoretical justification for why surrogate features effectively replace the original prompted features, relying instead on empirical results.

2. Analysis of performance sensitivity to hyperparameter choices, particularly for different loss components, the number of fine-grained text
features and multi-scale constant factor.

3. While the paper shows improved efficiency and competitive performance, there's limited analysis of whether the surrogate features might be less effective for certain types of tasks or datasets.

**Methods And Evaluation Criteria:**

Yes, the proposed methods and evaluation criteria are appropriate for the problem. The authors address the efficiency bottleneck in diverse prompt learning through a surrogate generation approach, which directly targets the computational overhead issue. The comparison metrics are relevant for assessing both performance and efficiency. The authors also appropriately evaluate their method against both single prompt learning approaches and diverse prompt learning approaches to demonstrate comprehensive improvements.

**Other Comments Or Suggestions:**

Nil.

**Other Strengths And Weaknesses:**

Nil.

**Questions For Authors:**

Nil.

**Relation To Broader Scientific Literature:**

Related to prompt engineering in VLM.

**Theoretical Claims:**

The paper does not contain formal mathematical proofs for theoretical claims. It presents algorithmic formulations and describes the approach using mathematical notation, but doesn't make rigorous theoretical claims requiring proof.

---

> ### Author Rebuttal · Authors · 2025-04-01
>
> >Q1: The paper doesn't provide much theoretical justification for why surrogate features effectively replace the original prompted features, relying instead on empirical results.
>
> A1: Thanks for the valuable comments. We provide a theoretical analysis to demonstrate the effectiveness of the surrogate features.
>
> Notation.
>
> Visual instance $x$, basic text prompt $s$, explicit diverse text prompt $s_E$, conditional signal $\alpha \in \mathbb{R}^d$, basic prompted text feature $w=T([s,c])\in \mathbb{R}^d$, surrogate prompted text feature: $h=\theta _ {SFG}(w,\alpha) \in \mathbb{R}^d$, the original explicit prompted text feature $w_E=T([s_E,c])\in \mathbb{R}^d$.
>
> Proof.
>
> To validate the effectiveness of replacing $w_E$ with $h$ in diverse prompt learning, we analyze their feature similarity through $||h-w_E||=||\theta _ {SFG}(w,\alpha)-w_E||$. Here, we mainly focus on providing the proof of surrogate instance-dependent prompt learning, which imposes stricter constraints (in surrogate fine-grained prompt learning, $\alpha$ is also learnable parameters, which directly simplified the condition to $h=\theta _ {SFG,\alpha} (w)$).
>
> $w$, $\alpha$, and $w_E$ are bounded-dimensional representations within the vector space $\mathbb{R}^d$ (outputs of continuous neural network mappings), and their corresponding inputs $s$, $x$ and $s_E$ are related via $s_E=s+\alpha=s+V(x)$ in instance dependent prompt learning (CoCoOp). Hence, there exists a continuous function $g$, such that: $w_E = g(w, \alpha)$. According to the universal approximation theorem, for any continuous function $g(\cdot)$ and any $\epsilon>0$, there exists a neural network with sufficient capacity (here, we utilize the cross-attention module $\theta_{SFG}$, which has been proved as a proper universal approximator$^\text{a}$) such that for all $(w,\alpha)$,
> $$
> || g(w,\alpha)-\theta_{SFG}(w,\alpha)||<\epsilon.
> $$
> By taking the approximation error $\epsilon \to 0$, we obtain $||h-w_E||\to 0$ at the optimal parameters $\theta_{SFG}^{\star}$, which confirms that $h$ effectively approximates and replaces $w_E$. This analysis validates the theoretical existence of $\theta_{SFG}^{\star}$, while extensive empirical studies in the manuscript further elaborate on its implementation and optimization, and substantiate its practical effectiveness.
>
> a. Are Transformers universal approximators of sequence-to-sequence functions? ICLR 2020.
>
> >Q2: Analysis of performance sensitivity to hyperparameter choices, particularly for different loss components, the number of fine-grained text features and multi-scale constant factor.
>
> A2: Thanks for the valuable suggestion. Here we carefully analyze the effect of different hyperparameters. Due to the character limit, we only report the averaged results on 11 datasets.
>
> Different loss components ($\lambda_1$ and $\lambda_2$): The results are quite stable under different loss coefficient hyperparameter settings. Since $\lambda_1=25$ and $\lambda_2=10$ achieve the best averaged performance, we choose them as the default setting.
>
> | $\lambda_1=15$ | $\lambda_1=20$ | $\lambda_1=25$ | $\lambda_1=30$ | $\lambda_1=35$ |
> |:------------:|:------------:|:------------:|:------------:|:------------:|
> |     85.02    |     85.04    |     85.12    |     85.09    |     85.06    |
>
> | $\lambda_2=5$ | $\lambda_2=10$ | $\lambda_2=15$ | $\lambda_2=20$ |
> |:-----------:|:------------:|:------------:|:------------:|
> |    84.93    |     85.12    |     85.12    |     85.09    |
>
> The number of fine-grained text features Z: Comparing the results between $Z=0$ (No FG prompts applied) and other settings validates involving FG information can significantly boost the performance. We observe consistent performance gains when increasing $Z$ from 1 to 4, which indicates applying sufficient FG prompt is necessary to capture comprehensive FG information. However, further increasing Z leads to performance degradation, which we attribute to the inclusion of ineffective information (e.g., background noise). Therefore, we select $Z=4$ as the default setting.
>
> | $Z=0$ | $Z=1$ | $Z=2$ | $Z=3$ | $Z=4$ | $Z=5$ | $Z=6$ |
> |:-----:|:-----:|:-----:|:-----:|:-----:|:-----:|:-----:|
> | 83.96 | 84.91 | 85.02 | 85.08 | 85.12 | 85.04 | 84.95 |
>
> Multi-scale constant factor $\eta$: In this paper, we utilize multi-scale strategy to obtain FG visual information under different scales. The results show that applying a relatively small $\eta$ (5 or 10) can achieve better performance, since applying over-large scale may involve ineffective information (e.g., background noise), thus leading to a negative effect.
>
> | $\eta=5$ | $\eta=10$ | $\eta=15$ | $\eta=20$ |
> |:--------:|:---------:|:---------:|:---------:|
> |   85.08  |   85.12   |   84.99   |   84.86   |
>
> >Q3: Limited analysis of whether the surrogate features might be less effective for certain types of tasks or datasets.
>
> A3: Thanks for the comments. Due to the character limits, please refer to Q&A 3 of Reviewer 4GML for details analysis.

---

### Official Review · Reviewer_Zno2 · 2025-03-11

**Overall Recommendation:** 3

**Summary:**

Prompt learning is an efficient fine-tuning technique that learns text prompts. Learning multiple text prompts instead of just one can improve performance while increasing computational cost. This paper proposes learning diverse text prompts without initializing additional parameters by generating specific text prompts with a lightweight model to avoid overwhelming the gradient process. The performance on several classification benchmarks is better than that of compared methods.

**Claims And Evidence:**

The claims in the submission are well-supported by clear evidence.

**Essential References Not Discussed:**

Nothing to supplement.

**Experimental Designs Or Analyses:**

The experimental designs and analyses are suitable and sufficient for this task.

**Methods And Evaluation Criteria:**

Yes.

**Other Comments Or Suggestions:**

None

**Other Strengths And Weaknesses:**

Strengths:
1. The method is simple and easy to follow.
2. The experiments are sufficient and show better speed compared to other diverse text prompt methods.

Weaknesses:
1. How does the FG loss work with images that contain multiple classes? Would other instances in the figure disturb the classification since we only need to predict the most significant object? For instance, the author could provide heatmap visualization on such images.
2. The description in Figure 2 is not clear enough, such as how the ID conditional signals interact inside the Surrogate Feature Generator. Moreover, the structure on the right does not match Eq(4), where ω∈M×d, α∈Z×d; then how does the output h_FG become Z×M×d? If the conditional signals interact individually, you should improve the figure to show it, or it could be confusing to the reader.
3. Since the fine-grained module improves performance according to the ablation study, has the author tried to adopt this idea to existing prompt learning techniques? I wonder why there is no such usage in the current community.

**Questions For Authors:**

I am interested in the initialization of conditional signals, why is the ID signal derived from the visual feature while the FG signal is randomly initialized?

**Relation To Broader Scientific Literature:**

This paper proposes a more efficient approach to diverse text prompt learning for image classification transfer tasks.

**Theoretical Claims:**

There are no theoretical claims for the proposed methods, such as how the fine-grained loss promotes global minimum results through spatial feature consistency. The provided theoretical explanation for the optimization stages appears to be a simple regularization loss that mitigates overfitting, offering no new insights.

---

> ### Author Rebuttal · Authors · 2025-03-31
>
> >Q1: How does the FG loss work with images that contain multiple classes? Would other instances in the figure disturb the classification since we only need to predict the most significant object? For instance, the author could provide heatmap visualization on such images.
>
> A1: Thanks for the insightful comments. Other instances will not disturb the classification process. In both diverse prompt learning and surrogate prompt learning, each input text description is comprised by the text prompts and the classname. Classnames are pre-defined and remain fixed during optimization, thereby providing consistent class-level semantic guidance. To this end, every FG prompted text feature embeds information that strongly associated with its corresponding class, which ensures the classification procedure not being disturbed by other instances in the image.
>
> To further validate this problem, we provide additional visualization results. We select images that contain at least two distinct object categories, and visualize the attention heat maps of different surrogate fine-grained text features on such images. The corresponding heat maps are shown in “[multiclass-heatmap.png](https://postimg.cc/sBK9jYJB)”, which demonstrate the effectiveness of FG prompted text features in our proposed method.
>
> >Q2: The description in Figure 2 is not clear enough, such as how the ID conditional signals interact inside the Surrogate Feature Generator. Moreover, the structure on the right does not match Eq(4), where $w \in M\times d$, $\alpha \in Z\times d$; then how does the output $h^{FG}$ become $Z\times M\times d$? If the conditional signals interact individually, you should improve the figure to show it, or it could be confusing to the reader.
>
> A2: Thanks for the valuable suggestion. We are sorry for the unclear description of Eq(4). The conditional signals interact individually with each basic prompted text feature $w_m$. We rewrite Eq(4) as:
>
> $$
> h_m = \theta _ {SFG}(w_m,\alpha).
> $$
>
> This equation clearly states that for the basic prompted text feature corresponds to $m$-th class $w_m$, we will generate $Z$ fine-grained surrogate prompted text features $h^{FG} _ {m} \in \mathbb{R} ^ {Z \times d}$. To this end, we have $h^{FG} \in \mathbb{R} ^ {Z\times M\times d}$ for total $M$ classes.
>
> We will correct the corresponding description in the next version of the manuscript. We also add the input and output notations of SFG in Fig.2 for clear description, which is shown in “[Revised-Fig2.png](https://postimg.cc/JD1zLdt8)”.
>
> >Q3: Since the fine-grained module improves performance according to the ablation study, has the author tried to adopt this idea to existing prompt learning techniques? I wonder why there is no such usage in the current community.
>
> A3: Thanks for your comments. First, the idea of fine-grained prompt learning has been utilized in several VLM-based prompt learning methods, which we have already cited in the manuscript (e.g., Chen et al., 2023, Lafon et al., 2024). However, these methods suffer from the huge computation complexity problem, hindering their practical application in real-world scenarios.
>
> Second, we claim that SurPL is adaptable and integrable to implement different diverse prompt learning ideas based on any single prompt learning approach. To this end, we have tried to implement SurPL on different existing single prompt learning methods, including CoOp, MaPLe and PSRC. Specifically, we consider these methods as baseline and additionally introduce both surrogate instance-dependent and fine-grained text features on each method. The comparison results are actually reported in Appendix Sec.E Table.12, and demonstrate that utilizing both instance-dependent and fine-grained can significantly improve the performance of existing single prompt learning methods.
>
> To explicitly explore the effectiveness of the fine-grained idea, here we further conduct the following experiment. We consider CoOp, MaPLe and PSRC as baseline approaches, and exploit our method to only generate surrogate fine-grained features based on each approach. We provide the comparison of average performances on 11 datasets below.
>
> |       | Method | Method+FG | $\Delta$ |
> |-------|:------:|:---------:|:--------:|
> |  CoOp |  79.89 |   83.15   |   3.26   |
> | MaPLe |  81.79 |   83.80   |   2.01   |
> |  PSRC |  82.87 |   83.33   |   0.46   |
>
> >Q4: Why is the ID signal derived from the visual feature while the FG signal is randomly initialized?
>
> A4: Thanks for the question. For existing ID prompt learning methods (such as CoCoOp), text prompts are not randomly initialized for optimization, but require visual information as prior knowledge. So, the ID signals are derived from the visual feature, thus providing visual guidance. For existing FG prompt learning methods (such as PLOT and GalLoP), text prompts are randomly initialized and supposed to capture fine-grained information during optimization. Therefore, we also keep FG signals randomly initialized.

---

### Official Review · Reviewer_c1YQ · 2025-03-12

**Overall Recommendation:** 3

**Summary:**

This paper proposes Surrogate Prompt Learning (SurPL), a new approach to enhance the efficiency and diversity of prompt learning for VLMs. Instead of learning multiple diverse prompts, SurPL directly generates surrogate prompted text features through a lightweight Surrogate Feature Generator (SFG), reducing computational overhead while maintaining diversity.

The key contributions of this paper include: 1) A novel method that efficiently generates diverse text features instead of learning multiple prompts. 2) A cross-attention-based module that produces instance-dependent and fine-grained surrogate text features. 3) Extensive tests on 15 vision classification datasets show that SurPL achieves comparable or superior performance to existing diverse prompt learning methods, while significantly improving computational efficiency.4) SurPL-G, an extension of SurPL with Sharpness-aware Minimization (SAM), further enhances generalization across base-to-novel, cross-dataset, and cross-domain scenarios.

Overall, this paper introduces an efficient and effective approach to prompt learning, addressing the computational cost issue while maintaining strong adaptation ability in VLMs.

## update after rebuttal
I appreciate the author's efforts during the rebuttal period and am willing to maintain my original score.

**Claims And Evidence:**

The paper presents strong empirical results, but some claims need further support:

1. The claim that SurPL reduces complexity to O(M) is based on experiments only, without a formal proof. A theoretical complexity analysis would make this more convincing.

2. The surrogate feature generator (SFG) is effective, but there is no comparison with alternative feature generation methods.

3. SurPL is only tested on classification tasks, making its generalization to other vision-language tasks unclear.

**Essential References Not Discussed:**

The paper covers prompt learning literature well, but some essential references are missing （see previous section for details）:

1) Comparison with efficient fine-tuning methods

2) Computational efficiency research

3) Prompt learning in broader tasks

**Experimental Designs Or Analyses:**

The experimental setup is well-structured, but some aspects could be improved:

1) The experiments focus only on classification, limiting the assessment of SurPL’s generalization. Testing on VQA, object detection, or video understanding would provide a broader evaluation.


2) While the paper compares against diverse prompt learning methods, it lacks comparisons with non-prompt-based methods (e.g., Adapters, LoRA). Including these baselines would clarify whether prompt learning is the best approach.


3) Key parameters (e.g., Z, η) are fixed without sensitivity analysis. Investigating their impact would strengthen the robustness of conclusions.
Overall, the experiments are well-executed, but broader evaluations and deeper analyses would improve their reliability.

**Methods And Evaluation Criteria:**

The proposed SurPL framework is well-motivated, and the SFG is a reasonable design. However, some aspects could be improved:

1)  The method is only tested on classification tasks, limiting its applicability. Evaluating SurPL on VQA, object detection, or video understanding would better assess its generalization.

2) The paper compares SurPL with diverse prompt learning methods but lacks comparisons with non-prompt-based fine-tuning approaches (e.g., Adapters, LoRA). Adding such baselines would clarify whether prompt learning is the best approach for efficiency.

3) The claim of O(M) complexity is reasonable, but no detailed breakdown is provided. A formal complexity analysis would strengthen this argument.

**Other Comments Or Suggestions:**

The writing is generally clear, but a few areas could be improved for better readability and precision. Below are some minor suggestions:

- The introduction could better emphasize the novelty of SurPL compared to previous diverse prompt learning methods. While the paper discusses existing works, a more explicit contrast with traditional prompt learning and fine-tuning methods would help highlight its contributions.

- Some notations and explanations in the methodology section could be clarified. For example, the role of Z (fine-grained feature count) and η (multi-scale factor) is not well explained, and a brief intuitive description would improve clarity.

- The figures are informative, but adding a computational flow diagram for SurPL compared to traditional prompt learning would make the efficiency argument clearer.

- There are some minor typos and grammar inconsistencies, particularly in the experimental section. Careful proofreading would improve the overall presentation.

- The appendix could include more details on hyperparameter settings, optimizer choices, and training dynamics to enhance reproducibility.

**Other Strengths And Weaknesses:**

Strengths

1. The paper proposes an innovative method (SurPL) that improves prompt learning efficiency while maintaining diversity. The Surrogate Feature Generator (SFG) effectively reduces computational overhead, avoiding the high cost of traditional diverse prompt learning.

2. The experimental results are strong, tested on 15 datasets, demonstrating that SurPL significantly improves computational efficiency while achieving comparable or superior performance to existing state-of-the-art methods.

3. The paper is well-structured, with a clear motivation, well-designed experiments, and strong baseline comparisons. The methodology and experimental sections are logically presented and easy to follow.

Weaknesses

1.The O(M) computational complexity claim lacks a formal derivation and is currently supported only by empirical results. A complexity analysis would enhance the theoretical rigor of the argument.

2. The method is tested only on classification tasks, making it unclear whether SurPL generalizes to VQA, object detection, and multimodal reasoning. Expanding the evaluation to broader tasks would verify its generalizability.

3. The paper does not compare SurPL with non-prompt-based fine-tuning methods such as Adapters, LoRA, and BitFit, which are widely used for efficient VLM adaptation. Including such comparisons would clarify whether SurPL is the most effective approach for efficient model adaptation.

**Questions For Authors:**

1. The paper claims that SurPL reduces complexity to O(M) compared to O(BM) or O(ZM), but there is no formal derivation. Could you provide a step-by-step complexity breakdown?

2. SurPL is compared against diverse prompt learning methods, but not with non-prompt-based adaptation techniques like Adapters, LoRA, or BitFit. Why were these omitted?

3. The paper focuses only on classification tasks. Have you tested SurPL on VQA, object detection, or multimodal reasoning?

4. Some hyperparameters, like Z (fine-grained feature count) and η (multi-scale factor), are fixed in the experiments. How sensitive is SurPL’s performance to these choices?

5. SFG generates surrogate features efficiently, but why was cross-attention chosen over other feature generation techniques (e.g., Transformer-based mechanisms)?

**Relation To Broader Scientific Literature:**

The paper situates SurPL well within the literature on prompt learning for vision-language models, referencing key works on single and diverse prompt learning. However, there are some missing discussions:

1. The paper does not compare SurPL with Adapters, LoRA, or other efficient fine-tuning methods, which are widely used alternatives to prompt learning.

2. While the paper focuses on reducing computation, it does not connect with broader studies on efficient deep learning, such as low-rank adaptations, pruning, or quantization techniques.

3. Most cited works focus on classification tasks, but SurPL's relevance to VQA, object detection, or multimodal reasoning is not discussed.

**Theoretical Claims:**

The paper does not provide formal theoretical proofs for its main claims, particularly the O(M) computational complexity of SurPL. The argument is mainly supported by empirical results, which are convincing but not mathematically rigorous.

---

> ### Author Rebuttal · Authors · 2025-03-31
>
> >Q1: The O(M) computational complexity claim of SurPL lacks a formal derivation.
>
> A1: Thanks for the valuable comments. We provide a theoretical analysis of computational complexity here.
>
> Notations.
>
> Loss $L$, classnames ${c}=(c_m) _ {m=1}^M$, text encoder parameter: ${T}=({T}^{k}) _ {k=1}^K$, SFG parameter: $\theta_{SFG}$, text prompts parameter: $s=(s^k) _{k=1}^K$, encoder layer depth: $K$, the parameter size of SFG is much smaller than that of text encoder: $\theta _ {SFG}<<T$.
>
> Derivation.
>
> Optimizing the text prompt $s^1$ at the text encoder input position requires computing the gradient through the entire model. For each single surrogate output text feature, the back-propagation computation of SurPL can be written as:
> $$
> \frac{\partial L}{\partial s^1}=\underbrace{\frac{\partial L}{\partial \theta _ {SFG}(T^K([s^{K},c _ m^k]);\alpha_j)}\cdot \frac{\partial \theta _ {SFG}(T^K([s^{K},c _ m^k]);\alpha _ j)}{\partial T^K([s^K,c _ m^K])}} _ {\text{SFG}}\cdot \underbrace{\frac{\partial T^K([s^K,c _ m^K])}{\partial T^{K-1}([{s}^{K-1},c _ m^{K-1}])}\cdots \frac{\partial {T}^{1}([s^1,c _ m^1])}{\partial s^1}} _ {\text{Text Encoder}}.
> $$
> This gradient computation can be conceptually divided into two parts, which correspond to the text encoder and SFG. We denote the computational complexity for each output text feature as $O_T$ and $O_{SFG}$, respectively. As shown in Fig.1 (c), SurPL simultaneously involves $M$ text features back-propagation w.r.t text encoder, which equals to the complexity in single prompt learning: $O(M) = M \cdot O_T$. For the SFG stage, $(B+Z)M$ text features are involved in computation. The overall complexity of SurPL can thus be expressed as:
> $$
> O_{SurPL} = M \cdot O _ T + (B+Z)M \cdot O _ {SFG}
> $$
> Since $\theta_{SFG}<<T$, it follows that $O_{SFG}<<O_T$. Therefore, the second term is negligible, and we can approximate: $O _ {SurPL} \approx M \cdot O_T = O(M)$.
>
> >Q2: Comparison with non-prompt-based PEFT methods.
>
> A2: Thanks for the suggestion. We further compare SurPL with Adapters (CLIP-Adapter(Gao et al., 2024), Tip-Adapter(Zhang et al., 2022), TaskRes(Yu et al., 2023)), BitFit (CLIPFit$^\text{a}$) and LoRA (CLIP-LoRA$^\text{b}$) based methods. The averaged performances of 11 datasets are shown below, which demonstrate the superiority of SurPL compared to other PEFT techniques. We will add more detailed results and analysis in the next version of the manuscript.
>
> | CLIP-Adapter | Tip-Adapter | TaskRes | CLIPFit | CLIP-LoRA |  DVLP | SurPL |
> |:------------:|:-----------:|:-------:|:-------:|:---------:|:-----:|:-----:|
> |     79.86    |    81.15    |  80.75  |  81.27  |   82.95   | 82.92 | 85.12 |
>
> a. Vision-Language Model Fine-Tuning via Simple Parameter-Efficient Modification. EMNLP 2024.
>
> b. Low-Rank Few-Shot Adaptation of Vision-Language Models. CVPRW 2024.
>
> >Q3: Experiments focus only on classification. Testing on VQA, object detection, or video understanding would provide a broader evaluation.
>
> A3: Thanks for your valuable suggestion. First, to our best knowledge, almost all studies of VLM-based PEFT techniques (e.g. prompts, adapters, LoRA, BitFit) mainly evaluate on classification tasks, and we simply follow these established protocols for fair comparison.
>
> Second, although some researches have applied PEFT techniques on other visual tasks, they tend to leverage existing methods rather than explore new approaches. Diverse prompt learning has not been well-explored on other tasks, making it challenging to identify a suitable baseline for SurPL on such tasks. Additionally, exploring the effectiveness of diverse prompt learning on VQA and detection from scratch and then implementing SurPL require significant time, making it difficult to include such results within the rebuttal period.
>
> However, we sincerely appreciate this insightful suggestion and will actively explore this direction in the near future.
>
> >Q4: Hyperparameter sensitivity analysis is missing.
>
> A4: Thanks for the comments. Due to the character limit, we provide comprehensive hyperparameter analysis at the Q&A 2 of Reviewer jTad. The results indicate that the hyperparameters applied in this work are reasonable and relatively stable. Please refer to that response for details.
>
> >Q5: Why was cross-attention chosen as SFG over other feature generation techniques (e.g., Transformer-based mechanisms)?
>
> A5: Thanks for the comments. SFG aims to take basic prompted text feature and conditional signal as input, and generate the surrogate prompted feature according to the given signal. Cross-attention module is the most intuitive choice for this requirement. While Transformer-based mechanisms can also achieve it, stacking multiple attention blocks would significantly decrease the efficiency.
>
> >Q6: Minor suggestions of writing.
>
> A6: Thanks for the valuable suggestions. Due to the character limit, we are not able to provide detailed modifications in the rebuttal. We will revise them in the next version of the manuscript.

---

### Decision · Program_Chairs · 2025-05-01

**Decision:**

Accept (poster)

**Comment:**

The paper proposed a new prompt learning method called Surrogate Prompt Learning (SurPL) for adapting CLIP in downstream datasets. The main idea of SurPL is to learn a feature generator to generate diverse text features from a single prompt. The advantage of this framework over existing methods is that it does not require complex gradient computations while maintaining competitive performance. The paper received four reviews with 4x weak accept ratings and the reviews are mostly positive. Specifically, the reviewers appreciated the extensiveness of the results and the strong performance. On the negative side, the reviewers mentioned that the method lacks a theory that justifies why the complexity is low compared to existing methods. Currently only empirical results are provided. Considering that the paper has solid results and the method has novelty, the AC recommends that the paper be accepted.